# Global tropospheric ozone trends, attributions, and radiative impacts in 1995–2017: an integrated analysis using aircraft (IAGOS) observations, ozonesonde, and multi-decadal chemical model simulations

Haolin Wang[1,2], Xiao Lu[1,2], Daniel J. Jacob[3], Owen R. Cooper[4,5], Kai-Lan Chang[4,5], Ke Li[6], Meng Gao[7], Yiming Liu[1,2], Bosi Sheng[1], Kai Wu[8], Tongwen Wu[9,10], Jie Zhang[9,10], Bastien Sauvage[11], Philippe Nédélec[11], Romain Blot[11], and Shaojia Fan[1,2]

[1] School of Atmospheric Sciences, Sun Yat-sen University, and Key Laboratory of Tropical Atmosphere-Ocean System, Ministry of Education, Zhuhai, China.
[2] Guangdong Provincial Observation and Research Station for Climate Environment and Air Quality Change in the Pearl River Estuary, Southern Marine Science and Engineering Guangdong Laboratory (Zhuhai), Zhuhai, China.
[3] Harvard John A. Paulson School of Engineering and Applied Sciences, Harvard University, Cambridge, MA, USA
[4] Cooperative Institute for Research in Environmental Sciences, University of Colorado, Boulder, CO, USA.
[5] NOAA Chemical Sciences Laboratory, Boulder, CO, USA.
[6] Jiangsu Key Laboratory of Atmospheric Environment Monitoring and Pollution Control, Collaborative Innovation Center of Atmospheric Environment and Equipment Technology, School of Environmental Science and Engineering, Nanjing University of Information Science and Technology, Nanjing, China
[7] Department of Geography, State Key Laboratory of Environmental and Biological Analysis, Hong Kong Baptist University, Hong Kong, China
[8] Department of Land, Air, and Water Resources, University of California, Davis, CA, USA.
[9] CMA Earth System Modeling and Prediction Centre, Beijing, China
[10] State Key Laboratory of Severe Weather, Beijing, China
[11] Laboratoire d'Aérologie (LAERO), Université Toulouse III - Paul Sabatier, CNRS, Toulouse, France

*Correspondence to*: Xiao Lu (luxiao25@mail.sysu.edu.cn) and Shaojia Fan (eesfsj@mail.sysu.edu.cn)

**Abstract.** Quantification and attribution of long-term tropospheric ozone trends are critical for understanding the impact of human activity and climate change on atmospheric chemistry, but are also challenged by the limited coverage of long-term ozone observations in the free troposphere where ozone has higher production efficiency and radiative potential compared to that at the surface. In this study, we examine observed tropospheric ozone trends, their attributions, and radiative impacts from 1995–2017 using aircraft observations from the In-Service Aircraft for a Global Observing System database (IAGOS), ozonesondes, and a multi-decadal GEOS-Chem chemical model simulation. IAGOS observations above 11 regions in the Northern Hemisphere and 19 of 27 global ozonesonde sites have measured increases in tropospheric ozone (950-250hPa) by $2.7 \pm 1.7$ and $1.9 \pm 1.7$ ppbv decade$^{-1}$ on average, respectively, with particularly large increases in the lower troposphere (950-800 hPa) above East Asia, Persian Gulf, India, northern South America, Gulf of Guinea, and Malaysia/Indonesia by 2.8 to 10.6 ppbv decade$^{-1}$. The GEOS-Chem simulation driven by reanalysis meteorological fields and the most up-to-date year-specific anthropogenic emission inventory reproduces the overall pattern of observed tropospheric ozone trends, including the large ozone increases over the tropics of 2.1-2.9 ppbv decade$^{-1}$ and above East Asia of 0.5-1.8 ppbv decade$^{-1}$, and the weak

tropospheric ozone trends above North America, Europe, and high-latitudes in both hemispheres, but trends are underestimated compared to observations. GEOS-Chem estimates an increasing trend of 0.4 Tg year$^{-1}$ of the tropospheric ozone burden in 1995–2017. We suggest that uncertainties in the anthropogenic emission inventory in the early years of the simulation (e.g., 1995–1999) over developing regions may contribute to GEOS-Chem's underestimation of tropospheric ozone trends. GEOS-Chem sensitivity simulations show that changes in global anthropogenic emission patterns, including the equatorward redistribution of surface emissions and the rapid increases in aircraft emissions, are the dominant factors contributing to tropospheric ozone trends by 0.5 Tg year$^{-1}$. In particular, we highlight the disproportionately large, but previously underappreciated, contribution of aircraft emissions to tropospheric ozone trends by 0.3 Tg year$^{-1}$, mainly due to aircraft emitting $NO_x$ in the mid- and upper troposphere where ozone production efficiency is high. Decreases in lower stratospheric ozone and the stratosphere-troposphere flux in 1995–2017 contribute to an ozone decrease at mid- and high-latitudes. We estimate the change in tropospheric ozone radiative impacts from 1995–1999 to 2013–2017 is +18.5 mW m$^{-2}$, with 43.5 mW m$^{-2}$ contributed by anthropogenic emission changes (20.5 mW m$^{-2}$ alone by aircraft emissions), highlighting that the equatorward redistribution of emissions to areas with strong convection and the increase in aircraft emissions are effective for increasing tropospheric ozone's greenhouse effect.

# 1 Introduction

Tropospheric ozone is a major air pollutant that has detrimental effect on human physiology and ecosystem productivity, and controls the oxidizing capacity of the troposphere as the dominant source of hydroxyl radicals (OH) (Atkinson, 2000; Jacob, 2000; Monks et al., 2015; Fleming et al., 2018; Unger et al., 2020). It is also a short-lived climate forcer interacting with both solar (shortwave, SW) and terrestrial (longwave, LW) radiation (IPCC, 2021). Tropospheric ozone is produced chemically from anthropogenic and natural precursors, it is also transported from the stratosphere, and is removed by chemical loss and dry deposition. The ozone lifetime spans from hours in the polluted boundary layer to a few weeks in the free troposphere, sufficiently short that ozone distributions and trends are highly variable. Chemistry climate models indicate an increase in the tropospheric ozone burden since the 1990s (Skeie et al., 2020), and regional changes in tropospheric ozone levels are likely to be caused by shifts in anthropogenic emissions of ozone precursors (Zhang et al., 2016) and climate (Lin et al., 2014; Lu et al., 2019b). Chemical models have been extensively used for quantifying the drivers of ozone trends at individual sites or regions and for estimating ozone radiative impacts, but their applications to the continental and global scales are largely constrained by the limited coverage of robust long-term ozone measurements for evaluating modelled ozone trends, especially in the free troposphere (Gaudel et al., 2020) where ozone has greater radiative impacts than at the surface (Lacis et al., 1990; Hansen et al., 1997). Here, we integrate long-term aircraft ozone observations, ozonesonde measurements, and multi-decadal chemical model simulations to quantify global tropospheric ozone trends, their attributions, and the resulting radiative impacts for 1995–2017.

The first phase of the Tropospheric Ozone Assessment Report (TOAR-I) initiated in 2014 utilized available surface, ozonesonde, aircraft, and satellite observations to assess tropospheric ozone trends from 1970 to 2014 (Schultz et al., 2017). TOAR-I concluded that observations through 2014 were not sufficient to detect an unambiguous trend in global tropospheric ozone burden over the past two decades (Gaudel et al., 2018). The Intergovernmental Panel on Climate Change Sixth Assessment Report [IPCC; AR6; section 2.2.5.3] assessed the historical ozone records from the early and mid-20[th] century to present-day (Gulev et al., 2021), concluding that "Based on sparse historical surface/low altitude data tropospheric ozone has increased since the mid-20th century by 30–70% across the NH (medium confidence). Surface/low altitude ozone trends since the mid-1990s are variable at northern mid-latitudes, but positive in the tropics [2 to 17% per decade] (high confidence). Since the mid-1990s, free tropospheric ozone has increased by 2–7% per decade in most regions of the northern mid-latitudes, and 2–12% per decade in the sampled regions of the northern and southern tropics (high confidence). Limited coverage by surface observations precludes identification of zonal trends in the SH, while observations of tropospheric column ozone indicate increases of less than 5% per decade at southern mid-latitudes (medium confidence)." An updated assessment from Cooper et al. (2020) reported a range of positive and negative ground-level ozone trends from 1995 to 2018 at 27 globally distributed remote sites. However, these sites represent less than 25% of the global surface area and are not indicative of the free troposphere. Ozonesondes provide measurements in the free troposphere, but their representativeness of regional trends is

largely limited by the low sampling frequency (2-3 times per week or lower) and their sparse spatial coverage on the continental scale (Tarasick et al., 2019; Chang et al., 2022). Satellite instruments provide high-resolution observations of tropospheric column ozone with excellent spatial coverage. Long-term tropospheric ozone trends from limited satellite products indicate increases across the tropics since the 1980s and 1990s (Ziemke et al., 2019; Gulev et al., 2021), but satellite-detected trends at mid-latitudes since the early 2000s are less certain due to instrument errors (e.g., row anomaly), uncertainties in retrieval algorithms, and disagreements between the available products (Gaudel et al., 2018). Ozone measurements from the In-Service Aircraft for a Global Observing System database (IAGOS) (Petzold et al., 2015), which contains ozone profiles from more than 60,000 commercial aircraft flights worldwide since 1994, are a critical source of data for quantifying ozone trends in the free troposphere (Cohen et al., 2018). A recent study utilizing IAGOS observations identified remarkable ozone increases in the free troposphere since 1994 above multiple regions of the Northern Hemisphere (Gaudel et al., 2020). The IAGOS observations provide a new opportunity for checking the consistency of ozone trends derived from other observation platforms (e.g., ozonesondes), and for evaluating the performance of chemical models used for interpreting ozone trend attributions and radiative impacts.

Modeling studies attributing long-term ozone trends have largely concentrated on the ground level, and on individual sites or regions of the U.S. and Europe where extensive surface observations are available (Parrish et al., 2014; Yang et al., 2014; Lin et al., 2017; Lu et al., 2018; Xu et al., 2018; Yan et al., 2018), while analyses of tropospheric ozone trend attribution on larger scales are rather limited. Zhang et al. (2016) showed that increases in the simulated global tropospheric ozone burden between 1980 and 2010 were dominated by the equatorward redistribution of anthropogenic emissions to developing regions in the tropics, particularly in East and South Asia. The rise in tropospheric ozone burden over the past two decades was also reproduced in models from the Phase 6 of the Coupled Model Intercomparison Project (CMIP6) (Griffiths et al., 2021), but quantification of ozone trend drivers was not available. Lu et al. (2019b) revealed that the ozone increases in the Southern Hemisphere troposphere over 1990–2010 were linked to the poleward expansion of the Hadley Circulation and associated changes in transport patterns and ozone production efficiency. The sparsity of available concurrent observations hindered a more comprehensive evaluation of simulated ozone trends in all of the above analyses and thus limits the interpretation of model results. We also lack an updated quantitative evaluation of anthropogenic and climatic drivers of global and continental ozone trends for years after 2010, when anthropogenic emissions of ozone precursors show contrasting changes compared to earlier years in regions such as China (Zheng et al., 2018). Estimates of ozone radiative impacts in previous studies are relative to the pre-industrial period (Stevenson et al., 2013; Skeie et al., 2020) and are poorly constrained by long-term ozone measurements in the troposphere, with no reliable ozone observations prior to the 20th Century (Tarasick et al., 2019).

In this study, we aim to update our understanding of tropospheric ozone trends, their attributions, and attendant radiative impacts on the global scale for the period 1995–2017 from chemical model simulations evaluated against extensive long-term

ozone measurements. We focus on the years 1995–2017 when ozone measurements in the free troposphere become increasingly available and state-of-the-science gridded anthropogenic emission inventories are reliable. We use the GEOS-Chem chemical transport model driven by assimilated meteorological fields and the most up-to-date global anthropogenic emission inventory to interpret tropospheric ozone trends for this period. We evaluate the GEOS-Chem tropospheric ozone trends with the aircraft ozone observations from the IAGOS database which provides long-term ozone measurements in

multiple regions, together with ozonesonde measurements; we also compare the GEOS-Chem results with CMIP6 models. We then conduct a series of sensitivity simulations to quantify the impact of anthropogenic emissions and climate change on global and continental tropospheric ozone trends and estimate the resulting radiative impacts.

## 2 Materials and Methods

### 2.1 IAGOS observations

The IAGOS program is a European Research Infrastructure (data available at https://www.iagos.org/, last access: March 9th, 2022) initiated in August 1994 (Thouret et al., 1998) that measures atmospheric composition worldwide using instruments onboard commercial aircraft of internationally operating airlines (Nédélec et al., 2015). Ozone is measured using a dual-beam ultraviolet absorption monitor with a time resolution of 4s, a precision of $\pm$ 2%, and an accuracy of about $\pm 2$ nmol mol$^{-1}$ (Thouret et al., 1998; Nédélec et al., 2015). IAGOS data have been regularly calibrated and show internal consistency for the

duration of the program (Blot et al., 2021). Measurements are made at any time of the day, during take-off and landing and during the cruise portion of the flight. The sampling frequency varies depending on the airline schedule but can be as high as four profiles per day in regions such as western Europe, enabling a robust estimate of free tropospheric ozone changes. Evaluations of IAGOS data show that they are consistent with ozonesonde records in the upper troposphere-lower stratosphere above western Europe (Staufer et al., 2013, 2014), and are representative of ozone in the lower troposphere (Petetin et al.,

2018; Cooper et al., 2020). Previous studies have applied IAGOS data to estimate regional-scale tropospheric ozone trends from the northern mid-latitudes to the tropics (Cohen et al., 2018; Cooper et al., 2020; Gaudel et al., 2020).

We focus on 11 regions with extensive IAGOS ozone profile sampling between 1995 and 2017, following Gaudel et al. (2020), as illustrated in Figure 1a, to estimate tropospheric ozone trends; to the best of our knowledge this is the first time that IAGOS data have been used to evaluate long-term (> 20 years) tropospheric ozone trends in a global chemistry transport model. These

11 study regions have frequent sampling in both the early (1995–2004) and late periods (2011–2017) between 1995 and 2017, allowing the estimation of tropospheric ozone trends over periods spanning two decades (see Section 2.5). As shown in Fig.1, western Europe is the region with the most frequent IAGOS ozone sampling with 33,563 available ozone profiles between 1995 and 2017, followed by Eastern North America (8281 profiles), East Asia (4192 profiles), Southeast US (4016 profiles), and Southeast Asia (2564 profiles). All regions except for Malaysia/Indonesia (567 flights) have more than 1000 ozone profiles

in this period. The inclusion of IAGOS data in Asia, Africa, and South America provides a unique opportunity to evaluate and interpret tropospheric ozone trends in these developing regions.

## 2.2 Ozonesonde observations

We also use ozonesonde measurements from the World Ozone and Ultraviolet Radiation Data Centre (WOUDC; available at https://woudc.org/data.php, last access: March 9th, 2022). WOUDC is operated by the Meteorological Service of Canada,

within Environment and Climate Change Canada. For our study period of 1995-2017, the database contains ozone profiles from 130 globally distributed sites. Ozone from the surface to the stratosphere (from launch up to 35 km) is measured by balloon-borne ozone electrochemical concentration cell instruments (Tarasick et al., 2019) with a vertical resolution of about 100 m and an accuracy of 5%-15% in the troposphere and 5% in the stratosphere (Sterling et al., 2018; Witte et al., 2018; Steinbrecht et al., 2021). The sampling frequencies of ozonesondes vary across sites but are mostly lower than 2-4 profiles a

week, posing a challenge for our ability to detect a trend. Chang et al. (2020) estimated that 18 profiles per month are required for accurate and robust long-term trend quantification at a single monitoring station. As very few monitoring locations can achieve such sampling frequencies, we soften the criteria to have (1) at least 3 observations per month for calculating the monthly mean; (2) at least 2 monthly observations for seasonal mean, and at least 8 months for annual mean; (3) at least 15 annual mean observations for the period of 1995–2017. Ozonesonde sites that meet these criteria and used in this study are

presented in Figure 1b and Table 1, including 18 and 9 ozonesonde sites in the Northern and Southern Hemispheres, respectively. These stations have been used to study tropospheric ozone trends in North America, Europe, Japan, and the Southern Hemisphere (Oltmans et al., 2013; Tarasick et al., 2016; Zeng et al., 2017; Lu et al., 2019b; Kumar et al., 2021; Chang et al., 2022).

## 2.3 GEOS-Chem model description and configuration

We apply the global three-dimensional chemical transport model GEOS-Chem version 13.3.1 (available at https://github.com/geoschem/GCClassic/tree/13.3.1, last access: March 9th, 2022, Bey et al. (2001)) to interpret global tropospheric ozone and its trends for 1995–2017. The model is driven by Modern Era Retrospective analysis for Research and Application version 2 (MERRA-2) assimilated meteorological fields from the NASA Global Modeling and Assimilation Office (GMAO), which has a native horizontal resolution of $0.5°$ (latitude) $\times 0.625°$ (longitude) and 72 vertical layers extending from

surface to 0.01 hPa (Gelaro et al., 2017).

GEOS-Chem describes coupled ozone–$NO_x$–VOCs–aerosol–halogen tropospheric (Wang et al., 1998; Park et al., 2004; Parrella et al., 2012; Mao et al., 2013) and stratospheric chemistry (Eastham et al., 2014). The model prescribes methane at the surface on the basis of spatially interpolated monthly mean surface methane observations from the NOAA Global Monitoring Division in 1995–2017, and allows its transport and chemistry (Murray, 2016). Gridded monthly surface mixing ratios for

N$_2$O, CFCs, HCFCs, halons, and organic chlorine species for 1995–2017 are obtained from the World Meteorological

Organization (Daniel et al., 2007), and are used as boundary conditions for these species in the model. The chemical kinetics

are from the Jet Propulsion Laboratory (JPL) and International Union of Pure and Applied Chemistry (IUPAC) (Sander et al.,

2011; IUPAC, 2013). Photolysis rates are calculated by the Fast-JX scheme (Bian and Prather, 2002). Advection of tracers in

GEOS-Chem is performed using the TPCORE advection algorithm (Lin and Rood, 1996). The boundary layer mixing process

is described by a non-local scheme (Lin and McElroy, 2010). Dry deposition of both gas and aerosols is calculated by a

resistance-in-series algorithm (Wesely, 1989; Zhang et al., 2001), with updates in ozone deposition to the ocean as described

by Pound et al. (2020). Wet deposition for water-soluble aerosols and gases is described by Liu et al. (2001) and Amos et al.

(2012).

Emissions in GEOS-Chem are operated by the Harvard-NASA Emission Component (HEMCO) (Keller et al., 2014). We

apply the latest version of the Community Emissions Data System inventory (CEDSv2), which builds on the extension of the

CEDS system to 2017 as described in McDuffie et al. (2020), to provide year-specific global anthropogenic emissions for

1995–2017 in GEOS-Chem (O'Rourke et al., 2021). The early version of this emission inventory, CEDS$_{CMIP6}$, provided gridded

(0.5° × 0.5°) monthly emissions of reactive gases and aerosols from 1750 to 2014 (Hoesly et al., 2018) and was used in the

CMIP6 experiment (Eyring et al., 2016; Hoesly et al., 2018). The CEDS$_{CMIP6}$ emissions from European countries, US, Canada,

and Australia are scaled to emissions from well-developed regional emission inventories, including the European Monitoring

and Evaluation Programme (EMEP) (EMEP, 2016), the US National Emissions Inventory (NEI) (US EPA, 2016), Canadian

Air Pollutant Emissions Inventory (APEI) (ECCC, 2016), and Australian National Pollutant Inventory (NPI) (ADE, 2016). As

for Asia, the CEDS$_{CMIP6}$ is scaled to an updated version of MIX inventory for China (Li et al., 2017) and the Regional Emissions

Inventory in Asia (REAS) for other Asia (Kurokawa et al., 2013). Here CEDSv2 updates activity data for combustion- and

process-level emission sources, and incorporates new regional inventories for India (Venkataraman et al., 2018) and Africa

(Marais and Wiedinmyer, 2016). In addition, CEDSv2 emissions are also scaled to the latest EMEP (EMEP, 2019), NEI (US

EPA, 2019), APEI (ECCC, 2019), NPI (ADE, 2019), and MEIC (Zheng et al., 2018) emission inventories for Europe, US,

Canada, Australia, and China, respectively, enabling the extension of emission estimates to 2017.

Figure 2 presents the evolution of global anthropogenic NO$_x$, CO, and NMVOCs emissions and the spatial distributions of

their trends in 1995–2017 from the CEDSv2 used in this study. Global anthropogenic NO$_x$ emissions showed increases from

34.5 TgN in 1995 to 40.5 TgN in 2008, then leveled off from 2008 to 2012, and decreased afterward to 36.2 TgN in 2017.

Global anthropogenic CO emissions decreased from 1995 of 648.6 Tg, leveled off between 2002 and 2012, and decreased

again afterward to 539 Tg in 2017. Global anthropogenic NMVOCs emissions showed increases from 133.8 Tg in 1995 to

149.3 Tg in 2012 and then flattened afterwards. Linear trends in global anthropogenic NO$_x$, CO, and NMVOCs emissions are

0.58 TgN year$^{-1}$, -2.7 Tg year$^{-1}$, and 0.96 Tg year$^{-1}$, respectively. Inspection of the regional trends shows that the global

anthropogenic emissions have shifted from developed regions in the northern mid-latitudes such as Europe and North America, where regulations of anthropogenic emissions were implemented in the 1990s (Archibald et al., 2017), to developing regions in the tropics and subtropics. The regions with the largest increases in anthropogenic emissions are East and South Asia, the Middle East, and Africa. The decline in global anthropogenic $NO_x$ and CO emissions after 2012 is largely driven by emission

reduction in China associated with the implementation of emission control strategies, while NMVOCs emissions are not effectively mitigated (Zheng et al., 2018).

Figures 2 and S1 also compare the anthropogenic emission trends from CEDSv2 and $CEDS_{CMIP6}$ for 1995–2014. We find that the anthropogenic $NO_x$ and CO emissions from CEDSv2 are lower than those in the $CEDS_{CMIP6}$ inventory in particular for years after 2007 by 8.8% and 3.9%, respectively. This leads to a much smaller trend of global anthropogenic $NO_x$ and CO in

CEDSv2 (0.58 TgN year$^{-1}$ and -2.7 Tg year$^{-1}$) compared to those in the $CEDS_{CMIP6}$ (1.1 TgN year$^{-1}$ and 0.32 Tg year$^{-1}$) for 1995-2014. Anthropogenic NMVOCs emissions are also smaller in the CEDSv2 than $CEDS_{CMIP6}$ (150.1 Tg versus 164.6 Tg in year 2014). These differences mainly reflect the updated regional inventory for China (Zheng et al., 2018), along with the inclusion of regional inventories for DICE-Africa (Marais and Wiedinmyer, 2016) and SMoG-India (Venkataraman et al., 2018), as well as the updated activity data in CEDSv2 (Figure S1).

The default GEOS-Chem model includes a monthly three-dimensional gridded inventory of aircraft emissions of $NO_x$, CO, and hydrocarbons based on the Aviation Emissions Inventory v2.0 (AEIC) for the year 2005, resulting in 0.96 TgN of global aircraft NO emissions with no interannual variability. A new study has showed that aircraft activity has exploded in recent decades, with aircraft $CO_2$ emissions 79.8% greater in 2018 relative to 1995 (Lee et al., 2021). Here we use the CEDS global aircraft emissions in 1995–2017 (O'Rourke et al., 2021), allowing our simulation to capture the impact from increases in

aircraft emissions on global ozone trends. The global aircraft emissions of $NO_x$, CO, and NMVOCs estimates for 2005 in the CEDS inventory are 0.88 TgN, 0.54 Tg, and 0.08 Tg respectively, slightly lower than those in the AEIC inventory. We find that the aircraft emissions of $NO_x$, CO, and NMVOCs increased from 1995 to 2017 by 0.51 TgN (76.2%), 0.26 Tg (58.1%), and 0.05 Tg (88.4%), respectively (Figure 2), consistent with Lee et al. (2021). Aircraft $NO_x$ emissions account for only 3.3% of the total anthropogenic emissions in 2017, however, as we will see later, they play an important role in the global

tropospheric ozone trends.

GEOS-Chem includes on-line calculation of biogenic emissions of NMVOCs, and $NO_x$ emissions from soil and lightning. Biogenic emissions are calculated using the Model of Emissions of Gases and Aerosols from Nature (MEGAN, version 2.1) (Guenther et al., 2012). Soil $NO_x$ emissions are calculated based on the availability of nitrogen (N) in the soil and edaphic conditions such as soil temperature and moisture (Hudman et al., 2010; Hudman et al., 2012; Lu et al., 2021). Lightning $NO_x$

emissions are parameterized as a function of cloud-top height (Price and Rind, 1992) and are then vertically distributed according to Ott et al. (2010). The spatial pattern of lighting $NO_x$ emissions is further constrained by climatological observations of lightning flash rates from the Lightning Imaging Sensor (LIS) and Optical Transient Detector (OTD) satellite instruments (Sauvage et al., 2007; Murray et al., 2012). Biomass burning emissions in 1995–2017 are from the BB4CMIP inventory as described in van Marle et al. (2017), in which the emissions for years after 1997 are the same as the Global Fire Emissions Database version 4 (GFED4; van der Werf et al. (2017)). Fire plumes can be injected beyond the planetary boundary layer (PBL). We partition 65% of the biomass burning emissions to the PBL and the remaining 35% into the free troposphere following Fischer et al. (2014) and Travis et al. (2016).

Model configurations are summarized in Table 2. We spin up the GEOS-Chem model by 10 years to provide an initial field for the atmospheric chemical components on 1 January 1995. The long spin-up time is to properly initialize the lower stratosphere. We conduct the standard simulation (BASE) from 1995 to 2017 using year-specific assimilated meteorology fields and anthropogenic and natural emissions as described above. We then conduct three sensitivity simulations to quantify the drivers of ozone trends. In the first sensitivity simulation FixAC, we fix global anthropogenic emissions (including aircraft emissions) and methane concentration at 1995 levels. However, mixing ratios of ozone depletion species (CFCs, HCFCs) are not fixed, as such their influences on the stratospheric ozone are available in the FixAC simulation. Ozone trends in the FixAC thus estimate the influence of climate (including their impacts on natural emissions) and stratospheric ozone on tropospheric ozone trends. The difference of ozone trends between the BASE and FixAC simulation then quantifies the contributions of anthropogenic emissions of tropospheric ozone precursors (including aircraft emissions and methane) to ozone trends. In the second simulation FixABC, we further fix biomass burning emissions at 1995 levels on the basis of FixAC, allowing us to examine the impact of biomass burning emissions alone on ozone trends. In the third simulation (FixAircraft), we fix global aircraft emissions at 1995 levels, and use the difference in ozone trend between BASE and FixAircraft to estimate the contribution of aircraft emissions alone to ozone trends.

We run the GEOS-Chem model at a horizontal resolution of 4° (latitude) × 5° (longitude), with 72 vertical layers extending from surface to 0.01 hPa. One-month model simulation at this resolution costs 4 hours with 48 CPUs (http://wiki.seas.harvard.edu/geos-chem/index.php/GEOS-Chem_13.3.0#1-month_benchmarks). Yielding 33-year (including 10-year spin-up simulation) global simulation of ozone trends thus require computation time of more than 60 natural days. As such we do not use a finer resolution of 2°× 2.5° that would otherwise cost at least eight times as much computational time and resources as in this study. This relatively coarse resolution of 4°× 5°may limit the ability of the model to capture finer-scale ozone trends, in particular at near surface where ozone and its precursor has a short lifetime. Artificial mixing of surface ozone precursors in coarse model grids may lead to higher-than-actual ozone production efficiency and therefore positive ozone biases which may further influence trend analyses (Wild and Prather, 2006; Yu et al., 2016; Young et al., 2018; Yin et

al., 2021). The limitation of model resolution, however, should be alleviated for ozone in the free troposphere, where ozone has longer chemical lifetime and should be better mixed than at near surface (Petetin et al., 2016). In light of this we do not use surface ozone observations for model evaluation, and mainly focus the trend analyses on above 950 hPa.

We also use model output from seven global climate-chemistry models in the CMIP6 historical experiments for comparison with the GEOS-Chem results and to examine the evolution of tropospheric ozone. An overview of the models included in this study is presented in Table 3. Model outputs are available on the Earth System Grid Federation (ESGF) website (https://esgf-node.llnl.gov/projects/cmip6/, last access: March 9th, 2022). All the CMIP6 historical simulations of ozone for 1850–2014 apply the $CEDS_{CMIP6}$ inventory as the global anthropogenic emissions of air pollutants (Feng et al., 2020) and the BB4CMIP6 inventory as biomass burning emissions, and apply the same external forcing from solar irradiance and well-mixed greenhouse gases (Meinshausen et al., 2017), but have significant differences in their resolutions, meteorology, chemical mechanisms, and representation of natural emissions such as lightning and biogenic emissions. The evolution of tropospheric ozone from 1850 to 2100 has been extensively analyzed in Griffiths et al. (2021). Here monthly model output of ozone for 1995–2014 is analyzed and compared with the GEOS-Chem results.

## 2.5 Trend estimation

We follow Gaudel et al. (2020) to determine tropospheric ozone trends from IAGOS and ozonesonde observations using the quantile regression method (Koenker and Bassett, 1978). The quantile regression method estimates trends based on the rank value of the sample distributions rather than the mean values, which makes no assumptions about the distribution of the data and has better tolerance to outliers (Koenker and Xiao, 2002; Chang et al., 2021). These advantages make it a robust tool for estimating trends of time series with many intermittent missing values, such as ozone records from the IAGOS and ozonesonde observations. More details of the method are described in Koenker and Hallock (2001).

We calculate ozone trends on 15 pressure levels at 50 hPa intervals from 950 to 200 hPa for each IAGOS region and each ozonesonde site. We remove data points with ozone higher than 125 ppbv at altitudes higher than 500 hPa to exclude the influence from episodic stratospheric intrusions and because the effect of these intrusions is diluted in the model (Zhang et al., 2014), based on observed ozone values in fresh stratospheric intrusions and in air pollution plumes (Nowak et al., 2004; Cooper et al., 2005; Archibald et al., 2020). Following Gaudel et al. (2020), we first calculate the monthly mean ozone values to construct the mean seasonal cycle from 1995 to 2017 for each layer and for each IAGOS region or ozonesonde site. The mean seasonal cycle is then used to deseasonalize each ozone record at the same pressure level. Finally, the quantile regression method is applied to calculate the linear trend of ozone using all available deseasonalized ozone profiles at each pressure layer. We report linear trends of ozone at the 50[th] (median) and 95[th] quantile in ppbv decade[-1] for the period 1995-2017 with a

corresponding *p*-value. Following the advice of the statistics community (Wasserstein et al., 2019) and as discussed in Gaudel et al. (2020) we do not use thresholds such as *p*-value < 0.05 to judge whether the reported trend is statistically significant.

## 2.6 Radiative impact calculations

We use the radiative kernel approach developed by Rap et al. (2015) to calculate the change in the radiative forcing of tropospheric ozone over the 1995-2017 period. The radiative kernel is defined as the derivative of the radiative flux relative to
a small perturbation in ozone. We use the radiative kernel from Skeie et al. (2020), which is constructed using the University of Reading (UoR) radiative transfer model (Myhre et al., 2011). UoR calculates ozone radiative forcing using the Edwards and Slingo (1996) two-stream radiation scheme that includes 8 bands in long-wave (Myhre and Stordal, 1997) and 6 bands in short-wave bands (Stamnes et al., 1988). Ozone radiative kernels have been widely used in previous studies to compare the radiative forcing of ozone across different chemistry-climate models (Rap et al., 2015; Iglesias-Suarez et al., 2018; Scott et al., 2018;
Skeie et al., 2020). Iglesias-Suarez et al. (2018) showed that ozone radiative forcing values calculated from the radiative kernel technique and from radiative transfer model are in good agreement with a global mean difference of 0.01 W m$^{-2}$. We interpolate the monthly ozone outputs from the GEOS-Chem simulations onto the T21 grid space (approximately 5.6° × 5.6°) and 60 vertical layers (ranging from the surface to 0.1 hPa) to match the resolution of the radiative kernel and then derive the radiative impacts.

**3 Results and Discussion**

### 3.1 Evaluation of GEOS-Chem tropospheric ozone

We evaluate the simulated tropospheric ozone and trends from the GEOS-Chem BASE simulation with the IAGOS and ozonesonde measurements. We sample the model outputs along the flight and sonde tracks and apply the same processes to simulated values as observations.

Figure 3 compares the annual vertical ozone profiles with the IAGOS observations over the 11 regions in the Northern Hemisphere for years 1995–1999 and 2013–2017. The model reproduces well the major features of tropospheric ozone vertical distributions, including the differences in the ozone increase with altitude between the northern mid-latitudes and tropics. The model shows good agreement with IAGOS observations in terms of the absolute ozone levels over Europe and North America.
Over East Asia, our GEOS-Chem simulation shows no significant ozone bias when averaging all IAGOS sampling data, but this reflects the offset between low bias in boreal spring and the high bias in summer. Park et al. (2021) also reported the ozone underestimation from eight chemical models including GEOS-Chem above South Korea during the Korea-United States Air Quality (KORUS-AQ) campaign in May–June 2016, and Gaubert et al. (2020) attributed this to missing CO sources in emission inventories for East Asia. The modelled ozone is biased high in the tropical regions particularly in boreal autumn and

winter (Table S1). We find that the GEOS-Chem ozone biases are smaller in 2013–2017 when activity data and emission factors are better constrained than in the early period of 1995–1999, smaller in regions where the CEDSv2 emission inventory is scaled to well-developed regional inventories (North America, Europe, East Asia) than in other regions, and larger in the lower troposphere than in the upper troposphere. In particular, the ozone low biases at 950-800 hPa layer above tropical Asia and Africa are much larger (11.3-15.9 ppbv) in 1995–1999 than in 2013–2017 (about 2.5 ppbv). A possible reason is that anthropogenic emissions in the early period and in developing tropical regions are biased high, which will also lead to underestimation of tropospheric ozone trends over these regions, as will be discussed later.

Figure 4 further compares the simulated vertical distributions of ozone with the ozonesonde measurements. We aggregate the ozonesonde data into six latitudinal bands for comparison, and results for individual sites are shown in Figure S2. The comparison again shows that GEOS-Chem captures the vertical structure of ozone at these globally distributed ozonesonde stations, but unlike the positive bias relative to IAGOS observations over industrialized areas, GEOS-Chem shows negative ozone biases in the free troposphere relative to ozonesonde measurement, in particular for remote sites in the extratropical regions by up to 20 ppbv. IAGOS and ozonesonde observations have very different spatial distributions (except overlaps in the Europe) (Fig.1) and reflect ozone difference over industrialized versus remote regions, so that inconsistency in simulated ozone bias can be expected. The low tropospheric ozone bias relative to ozonesonde observations in recent GEOS-Chem model versions has been demonstrated in several studies. The latest comprehensive evaluation of the global tropospheric ozone simulation using the version 10.1 of the model (Hu et al., 2017) found small low ozone biases compared to ozonesonde observations in the northern extratropical and polar regions, which were attributed to the underestimation of stratosphere-troposphere ozone exchange (STE) flux in that version of the model. The scientific updates since the version 10.1 (https://geos-chem.seas.harvard.edu/geos-new-developments), including the implementation of halogen (Cl-Br-I) chemistry in version 11-02 and 12.9 (Sherwen et al., 2016; Wang et al., 2021), update of heterogeneous $NO_y$ chemistry in aerosols and cloud in versions 12.6 (Holmes et al., 2019), and introduction of oceanic ozone deposition in version 12.8 (Pound et al., 2020) have significantly improved the model performance for many other chemical species but tend to enlarge the ozone low bias, in particular updated halogen chemistry further decreases surface ozone at high-latitude regions (Wang et al., 2021). Correcting the ozone low bias in remote regions in GEOS-Chem would be a topic of future research.

### 3.2 Tropospheric ozone trends from observations and chemical models

We estimate tropospheric ozone trends over 1995–2017 using IAGOS and ozonesonde observations (Figures 5-7 and Figure S3). Figures 5 and S3 show the vertical distributions of annual tropospheric ozone trends at the 50th and 95th percentiles from IAGOS observations, estimated using the quantile regression method as described in Section 2.5. IAGOS observations show that the 50th percentile of tropospheric ozone has increased over all 11 study regions in the Northern Hemisphere in 1995–2017, as also pointed out by Gaudel et al. (2020). Large ozone increases are found from IAGOS observations in the lower troposphere (950-800 hPa) for all seasons over developing regions in the northern tropics and subtropics, including East Asia,

Southeast Asia, Persian Gulf, India, northern South America, Gulf of Guinea, and Malaysia/Indonesia, with annual trends ranging from 2.8 to 10.6 ppbv decade$^{-1}$. The observed 95$^{th}$ percentiles of lower tropospheric ozone over these regions have increased by 3.6-24.2 ppbv decade$^{-1}$ (Figure S3), showing that the extreme ozone values are rising even faster. The positive trends extend to the free troposphere but with much smaller values. In comparison, the lower tropospheric trends of the 50$^{th}$ percentile ozone in developed regions (Europe, North America) over the northern mid-latitudes are much smaller by up to 1.8 ppbv decade$^{-1}$, which is largely driven by boreal autumn and winter with ~1.2 ppbv decade$^{-1}$ on average (Fig.S4). There are small negative trends in the annual 50$^{th}$ percentile in the lower troposphere above North America driven by ozone decreases in the summer (Fig.S4) (Cooper et al., 2012; Simon et al., 2015; Gaudel et al., 2020). The annual 95$^{th}$ percentile in the lower troposphere above Europe and North America has declined at the rate of -0.4~-8.3 ppbv decade$^{-1}$, which is consistent with surface ozone trends (Chang et al., 2017; Gaudel et al., 2018).

Figure 6 presents the vertical distributions of tropospheric ozone trends derived from ozonesonde observations, complementing ozone trend analyses from IAGOS by providing tropospheric ozone trends in remote regions and latitudes. We find the largest tropospheric ozone trends at the 50$^{th}$ percentile at three stations in the East Asia region, ranging from 3.8 to 6.7 ppbv decade$^{-1}$ throughout the troposphere. Ozone at these stations is affected by the outflow of Asian pollution plumes, and tropospheric ozone trends there are even larger than those over the source region estimated from IAGOS observations. Notable ozone increases are also found at two stations in the southern subtropics, La Réunion and Natal, with 0.04-6.1 ppbv decade$^{-1}$ in the middle troposphere and 4.6-9.3 ppbv decade$^{-1}$ in the upper troposphere, in agreement with the results of Witte et al. (2017). At stations with higher latitudes, tropospheric ozone trends are generally smaller and signs are varied among sites. As discussed in detail by Chang et al. (2020; 2022), low sampling frequencies at ozonesonde stations can make it difficult to detect a trend, and therefore discrepancies in trends from ozonesonde measurements in the free troposphere above Europe and western North America are not unexpected. When the sample size above these regions is maximized by combining all available ozonesonde and IAGOS profiles, the resulting combined product reveals increasing ozone above these regions from 1994 to 2019 (Chang et al., 2022).

We integrate in Figure 7a and Table 4 the annual trends in median ozone for tropospheric column (950-250 hPa) in 1995–2017 from both IAGOS and ozonesonde observations. This allows us to provide a more complete picture of observed global tropospheric ozone trends than previous studies focusing on the Northern Hemisphere alone (Gaudel et al., 2020), and to check the consistency between the two sources of ozone trend measurements. Both observational datasets consistently reveal widespread ozone increases in the troposphere over the past two decades, with larger ozone increases over developing regions in low latitudes than those over middle and high latitudes. All 11 IAGOS areas and 19 of 27 ozonesonde sites have measured increases in tropospheric ozone by 2.9 ± 1.7 and 1.9 ± 1.7 ppbv decade$^{-1}$ on average, respectively. In particular, trends in the northern low latitudes (0°-30°N) are 4.2 and 2.4 ppbv decade$^{-1}$ for ozone at the 50$^{th}$ percentile, averaged over all 6 IAGOS area

and 3 ozonesonde observations, respectively. In Europe and North America, observed trends are mostly positive, while three sites (Payerne, Legionowo, and Boulder) show inconsistently negative trends of -0.5~-0.6 ppbv decade$^{-1}$ that are in contrast to IAGOS observations (0.8~1.7 ppbv decade$^{-1}$) and trends at the other nearby sites (0.3-2.1 ppbv decade$^{-1}$). Increasing the sampling frequency (i.e. to 18 profiles month$^{-1}$ according to (Chang et al., 2020) would be helpful to reconcile the ozone trend estimate at adjacent ozonesonde sties, but we do not exclude the possibility that tropospheric ozone trends can still be variable even at adjacent locations. IAGOS observations provide less information in the Southern Hemisphere. Six of nine ozonesonde stations in the Southern Hemisphere show increasing tropospheric ozone, with the largest trends at low latitudes sites in La Réunion and Natal (4.6 ppbv decade$^{-1}$ and 3.0 ppbv decade$^{-1}$; *p*-value < 0.01), while ozone in the southern middle and high latitudes displays no significant tropospheric ozone trends in 1995–2017, with slight decreasing trends at two sites in Australia (Macquarie Island and Broadmeadows) and one site in Antarctica (Marambio).

Figures 5-7 and Table 4 also evaluate the performance of GEOS-Chem in reproducing the observed tropospheric ozone trends from IAGOS and ozonesondes. Our BASE simulation reproduces the overall pattern of tropospheric ozone trends in 1995–2017, in particular the larger ozone increases over the low latitudes, with a correlation coefficient *r*=0.6 for all pairs of observed and simulated tropospheric ozone trends. GEOS-Chem simulated trends in tropospheric ozone are 2.1-2.9 ppbv decade$^{-1}$ over East Asia, India, Southeast Asia, Persian Gulf, and Malaysia/Indonesia, accounting for 51.8-81.4% of the IAGOS trends over these rapidly developing regions (Table 4). We find a larger underestimation of ozone trends in the lower troposphere (950-800 hPa) compared to higher altitudes (Fig.5). The model also catches the positive tropospheric ozone trends at three ozonesonde stations in East Asia of 0.5-1.8 ppbv decade$^{-1}$, but are underestimated compared to the observed trends of 3.7-5.2 ppbv decade$^{-1}$. As discussed in Section 3.1, bias in anthropogenic emissions of ozone precursors during the early years of 1995–2017 over these developing regions may contribute to the underestimation of trends. The larger underestimation of tropospheric ozone trends over the Asian-Pacific ozonesonde sites may result from the coarse resolution of our simulation which is not adequate to resolve the Asian pollution outflow (Eastham and Jacob, 2017). In the northern middle and high latitudes (Europe and North America), GEOS-Chem estimates weak tropospheric ozone trends of -2.0 to 1.6 ppbv decade$^{-1}$ at IAGOS regions and ozonesonde stations, smaller than the spread of observed tropospheric ozone trends, reflecting the model difficulty in capturing weak and variable tropospheric ozone trends there. The model reproduces the ozone increases in the Southern low latitudes (0.14-0.94 ppbv decade$^{-1}$, compared to 1.7-4.6 ppbv decade$^{-1}$ from ozonesonde observations) except for Samoa and the varied tropospheric ozone trends over the Southern mid-latitudes and high-latitudes.

We compare the tropospheric ozone burden and trends in GEOS-Chem with the selected CMIP6 chemical models in Figure 8. Here our GEOS-Chem simulation serves as a platform to evaluate tropospheric ozone trends in CMIP6 models, as the monthly mean output of CMIP6 models hinders a direct comparison against IAGOS and ozonesonde observations. GEOS-Chem estimates a global tropospheric ozone burden of 304.9 Tg averaged over 1995–2014 and 311.1 Tg for year 2010, at the low

end of the eight CMIP6 models for 1995-2014 (308.1-347.5 Tg), and the IPCC AR6 multi-model ensemble and observational estimates for 2010 (347 ± 28 Tg) (Szopa et al., 2021), again reflecting the low ozone bias in current GEOS-Chem versions (Christiansen et al., 2022) and the lower emissions in our simulations than CMIP6 simulations (Fig. 2). The interannual variability of tropospheric ozone burden in GEOS-Chem is moderately consistent with the CMIP6 models with $r$ ranging from 0.3-0.6 (Fig. 8a).

All models show an increase in tropospheric ozone burden over the period 1995-2014, but the magnitude of trends differs by a factor of four. GEOS-Chem estimates an increasing trend in global tropospheric ozone burden of 0.2 Tg year$^{-1}$, which enlarges to 0.4 Tg year$^{-1}$ if 2015-2017 trends are included, but is still in the low end of the CMIP6 model ensemble (0.4 to 1.3 Tg year$^{-1}$). We find in Figures 8b and S5-S6 that all models agree with the significant ozone increases in 30°S-30°N, with tropospheric ozone burden increased by 2.4% in 2010–2014 (3.7% in 2013–2017) relative to 1995–1999 in GEOS-Chem and by 2.2%-8.0% in CMIP6 models, though the GEOS-Chem trends over 30°S-30°N are very likely underestimated compared to the observed trends as discussed above. However, GEOS-Chem simulation shows no notable ozone changes integrated in the 90-45°S and 45-90°N latitude bands, while a number of CMIP6 models show distinguished ozone increases. Our analyses of the observed tropospheric ozone trends from the IAGOS and ozonesonde observations in Figure 7 suggest some inconsistency in tropospheric ozone trends over the poleward 45° in the both hemispheres, indicating that the simulated ozone increases over these regions from some CMIP6 models need to be interpreted with caution. The weaker tropospheric ozone trends in our GEOS-Chem simulation compared to the CMIP6 models should mostly come from the smaller trends in global anthropogenic emissions in CEDSv2 compared to the CEDS$_{CMIP6}$ inventory, as discussed in Section 2.3, but may also reflect the differences in driven meteorology or model mechanism. Global tropospheric ozone seasonal trends and drivers will be discussed in the next section.

## 3.3 Factors contributing to the tropospheric ozone trend and burden increase from 1995 to 2017

We now examine the factors contributing to tropospheric ozone trends from 1995 to 2017 from GEOS-Chem sensitivity simulations. Figure 9 summarizes the contribution to tropospheric ozone burden trends. Figures 10-11 present the seasonal mean distributions of tropospheric ozone trends and contributions from different drivers, separating anthropogenic and climatic/stratospheric influences as described in Section 2.3. Figure S7 presents the ozone trends and attributions at the surface level.

We find that changes in global anthropogenic emissions, including surface emissions of short-lived ozone precursors, methane, and aircraft emissions, are the main drivers of the increase in global tropospheric ozone burden, and largely determine the overall spatial pattern of ozone trends in the BASE simulation in 1995–2017. Changes in anthropogenic emissions alone increase tropospheric ozone burden by 0.5 Tg year$^{-1}$ ($p$-values<0.01), compared to the total simulated tropospheric ozone

burden trend (0.4 Tg year$^{-1}$; $p$-values<0.05) in 1995-2017 (Fig.9). The emission-driven increases are particularly large (1.0 ppbv decade$^{-1}$) in the Northern low latitudes where we have observed the most notable ozone increases (Figs.10-11). We have shown in Figure 2 that the global anthropogenic emissions of NO$_x$ and NMVOCs emissions have been increasing and shifting equatorward from developed regions in the northern mid-latitudes in Europe and North America to the low-latitudes from 1995 to 2017, in particular anthropogenic emissions of NO$_x$ and NMVOCs have increased by 55.5% and 35.6%, respectively, in the 0°-30°N latitudinal band. Emissions of ozone precursors at low latitudes produce ozone at high efficiency due to the higher solar radiation, temperature, and NO$_x$ sensitivity compared to those at high latitudes (Zhang et al., 2016; 2021). Frequent deep convection at low latitudes also effectively lofts the pollutants to the upper troposphere and can further influence global tropospheric ozone trends via atmospheric circulation (Lawrence et al., 2003; Lu et al., 2019b). This is supported by the extended positive emission-driven ozone trends from the surface to upper troposphere over 0°-30°N (Fig. 10). Our result highlights the significant role of the emission-driven ozone increases in tropospheric ozone trends since 1995.

Changes in anthropogenic emissions contribute to tropospheric ozone increases in 30-90°N in spring, autumn, and winter, but lead to ozone decreases in summer (Fig.10). This pattern reflects the differences in emission-driven tropospheric ozone trends between East Asia (positive) and Europe and US (negative) (Fig.11). The CEDSv2 emission inventory (Figure 2) has documented rapid increases in anthropogenic emissions of ozone precursors over China, which contributed to significant increases in tropospheric ozone between 1995 and 2017 by about 2 ppbv decade$^{-1}$ for all seasons. We find that at the surface changes in the anthropogenic emissions alone lead to larger ozone increases in summer by 5 ppbv decade$^{-1}$ (Figure S7), but decrease in surface ozone by -4 ppbv decade$^{-1}$ in northern China in winter due to NO$_x$ titration effect However, we note that these surface trends are estimated from the coarse-resolution simulation at 4°× 5°. The emission-driven reduction of summertime ozone (and increases in wintertime ozone) over Europe and North America in 1995–2017 are clearly linked to the decline in anthropogenic NO$_x$ emissions as documented in the literature (Cooper et al., 2012; Simon et al., 2015; Lin et al., 2017; Gaudel et al., 2018; Yan et al., 2018).

Anthropogenic emissions also increase tropospheric ozone in the Southern Hemisphere by 0.5 ppbv decade$^{-1}$. Estimates from the CEDSv2 emission inventory in the Southern Hemisphere show that anthropogenic emissions of CO, NO$_x$ and NMVOCs increased by 26.5%-59.8% from 1995 to 2017. In addition, emission-driven ozone increases in the tropics would also extend to the Southern Hemisphere through meridional atmospheric circulation.

We highlight here the disproportionately large but previously underappreciated contribution of aircraft emissions to 1995–2017 tropospheric ozone trends. Our FixAircraft simulation allows us to separate the impact of aircraft emissions alone from the total anthropogenic emissions on ozone trends. As mentioned in Section 2.2, aircraft NO$_x$ emissions have increased by 76.2% from 1995 but still only account for 3.3% of the total anthropogenic NO$_x$ emissions for year 2017. However, they

contribute to a global tropospheric burden trend of 0.3 Tg year$^{-1}$ ($p$-values<0.01), accounting for 66% of the total emission-driven tropospheric ozone trends (Fig.9b). This disproportionately large contribution is because aircraft $NO_x$ emissions are mainly released in the middle and upper troposphere, where the $NO/NO_2$ ratio is high and the lifetime of $NO_x$ is long (Silvern et al., 2018), leading to a much higher ozone production efficiency compared to $NO_x$ emissions at the surface. We find that aircraft emission-driven trends are higher in the Northern Hemisphere than the Southern Hemisphere due to the greater density of flights, higher in December-January-February than in June-July-August because of longer lifetime of ozone and the lower

$NO_x$ levels in the free troposphere (as lightning $NO_x$ emissions are lower in boreal winter), and higher in the upper troposphere where (Fig.10) tropospheric ozone has the largest radiative impacts.

Climatic factors (including stratospheric influences and natural emissions, as diagnosed from the FixAC simulation) contribute little to the trend of the global tropospheric ozone burden (-0.1 Tg year$^{-1}$; $p$-values=0.3), but have significant influence on its

interannual variability. We find a high tropospheric ozone burden in 1998 and 2010 in GEOS-Chem and also in CMIP6 models. The high tropospheric ozone burden in these years is tied to the El Niño-Southern Oscillation (ENSO). ENSO influences global tropospheric ozone burden by modulating fire and lightning emissions, STE flux, and influences regional ozone by modulating the transport pattern and local weather relevant to the ozone photochemical environment (Zeng and Pyle, 2005; Lin et al., 2014; Lu et al., 2019a). In particular, biomass burning emissions of CO are 36.4% higher in 1998 (618.7 Tg) compared to

365.5 Tg averaged over 1995–2017, and we find that the anomalously high biomass burning emissions alone enhanced the tropospheric ozone burden by 7.8 Tg compared to a sensitivity simulation with fire activity fixed at the 1995 level. This is because the positive phase of ENSO (El Niño) in 1998 induces anomalous downward motion of air, which leads to hot and dry weather conditions over equatorial Asia and Central and South America that are favorable for strong fire activity (Doherty et al., 2006; van der Werf et al., 2008; Fonseca et al., 2017). This El Niño driven ozone peak in 1998 is more prominent in

GEOS-Chem than most of the CMIP6 models, very likely because El Niño driven shift in weather conditions and transport pattern is better reflected in MERRA2 re-analyses data used to drive our GEOS-Chem model, compared to climate fields simulated by CMIP6 climate-chemistry models without nudging to observed sea-surface temperature.

Figures 10-11 show that tropospheric ozone would decrease over mid-latitudes and high latitudes of both Hemispheres in the absence of anthropogenic emission changes in 1995–2017, as estimated from the FixAC simulation. We find significant ozone

decreases in the lower stratosphere in the Southern Hemisphere in March-August and in the Northern Hemisphere for all seasons, and the negative ozone trends extend downward to the troposphere (Fig.10), indicating that changes in stratospheric ozone and/or stratosphere-troposphere dynamics are contributing to tropospheric ozone decreases. The ozone decreases in the lower stratosphere are inconsistent among the CMIP6 models (Fig.S5). Our GEOS-Chem model by implementing the time-resolved surface concentrations of ozone-depleting-species as boundary conditions shows a moderate increasing trend in total

stratospheric ozone burden from 1995 to 2017 (Fig.S8), consistent with the observations suggesting a leveling off of declining

trends in stratospheric ozone after the Montreal Protocol (Solomon et al., 2016; Weber et al., 2022) and with other modeling studies (Griffiths et al., 2020). However, satellite observations have revealed that ozone in the lower stratosphere are still decreasing after the 1990s and the drivers are still not clear (Ball et al., 2020), supporting the negative ozone trends in the lower stratosphere in GEOS-Chem.

We further use two methods to diagnose the STE flux in the GEOS-Chem and examine their trends in 1995–2017. The first method diagnoses STE flux as a residual burden of ozone, calculated as STE = $O_3$ loss + $O_3$ dry dep – $O_3$ production + $\Delta O_3$, where $\Delta O_3$ is the change of tropospheric ozone burden relative to the previous year. This method is widely used in multi-model estimates of tropospheric ozone burden (Young et al., 2018; Archibald et al., 2020; Griffiths et al., 2020). The second method diagnoses the STE flux as the vertical ozone flux at 100 hPa (Hsu and Prather, 2014). As shown in Fig.S8, even though the

absolute values of STE flux are not consistent between the two methods, both methods suggest a negative trend in STE flux in 1995–2017, consistent with a recent study using reanalyses data to diagnose long-term trend in STE flux (Wang and Fu, 2021). The decrease in STE flux explains the tropospheric ozone decreases over the high-latitudes in the FixAC simulation. However, both methods are not applicable to derive STE trends at different latitude bands. More work is required to evaluate the trends in STE flux and to explore the driving factors.

**3.4 Radiative impacts of tropospheric ozone changes in 1995–2017**

We now examine the radiative impacts of tropospheric ozone changes in 1995–2017. Figure 12 shows the difference in modelled mean ozone in Dobson Units (DU) between 2013–2017 and 1995–1999. We use the five-year average for comparison to reduce the impact of short-term climate variability on ozone. Global average tropospheric column ozone increased by 0.6 DU in 2013–2017 compared to the 1995–1999 level, with the greatest increases in the tropics and in the upper troposphere

(Fig.12b), where ozone radiative impacts are the largest as reflected in the ozone radiative kernel (Fig.12c and Fig. S9).

Figures 13a-c estimate the associated changes in radiative forcing for total radiation (SW+LW) and separately for SW and LW, using the ozone radiative kernel method as described in Section 2.6. We find that the global averaged total tropospheric ozone radiative impact is 18.5 mW $m^{-2}$ in 2013–2017 compared to 1995–1999 level, with 1.6 mW $m^{-2}$ from the SW and 16.9 mW $m^{-2}$ from the LW. This estimated radiative impact value is approximately 4.7% of the tropospheric ozone radiative forcing of 390

(270 to 510) mW $m^{-2}$ in 2005–2014 relative to 1850 estimated from 10 climate models in CMIP6 using the same radiative kernel method (Skeie et al., 2020). In comparison, changes in global anthropogenic $NO_x$ emissions between 2017 and 1995 are 3.9% (1.7 TgN) of those between 2014 and 1850 (43.2 TgN). However, as our model has underestimated tropospheric ozone trends, the calculated ozone-induced radiative impacts are very likely smaller than the true forcing. Peak SW radiative impact is found in regions with large ozone changes and high albedo, such as over deserts or ice, and with low clouds. The

LW radiative impact peaks in areas with large ozone changes and with hot surface temperatures and high tropopause levels. Thus, we see large ozone radiative impact over the northern tropics.

Figures 13d-e further attribute the total tropospheric ozone radiative impacts to changes in anthropogenic emissions, aircraft emissions only, and to climatic and stratospheric influences. Changes in anthropogenic emissions contribute to tropospheric ozone radiative impact by 43.5 mW m$^{-2}$, representing the dominant factor driving the increase in tropospheric ozone radiative

impact from 1995. In particular, aircraft emissions alone contribute to tropospheric ozone radiative impact by 20.5 mW m$^{-2}$. The large emission-driven tropospheric ozone radiative impact increases from 1995 to 2017 are not only due to the increase in the absolute amount of emissions, but also reflect the equatorward redistribution of emissions to regions with strong convection and the increases in aircraft emissions, which have both led to ozone increases in the middle and upper troposphere and over the tropics, where the potential for tropospheric ozone radiative impacts are a magnitude of two larger than those at the surface

over mid-latitudes (Fig.12c). Nevertheless, our analysis does not reflect the long-term indirect radiative impacts of aircraft emissions through modulating tropospheric OH and CH$_4$ and stratospheric chemistry (Lee et al., 2021). The climatic and stratospheric influences contribute -25.1 mW m$^{-2}$ to tropospheric ozone radiative impact, mainly reflecting the simulated ozone decreases in the extratropical upper troposphere.

## 4 Conclusions

We examine the tropospheric ozone trends, their attributions, and radiative impact from 1995–2017 using aircraft (IAGOS) observations, ozonesondes, and a multi-decadal GEOS-Chem chemical model simulation. The combination of IAGOS and ozonesonde observations provides a global view of tropospheric ozone trends, and enables an extensive evaluation of GEOS-Chem simulated tropospheric ozone trends. We attribute tropospheric ozone trends to changes in anthropogenic emissions and climatic and stratospheric factors through a set of GEOS-Chem sensitivity experiments, and calculate the change in

tropospheric ozone radiative forcing during 1995–2017.

We find that both the IAGOS and ozonesonde observations reveal significant tropospheric ozone increases over the tropics in 1995–2017. The largest positive ozone trends in the lower troposphere are found over developing regions (East Asia, Persian Gulf, India, northern South America, Gulf of Guinea, and Malaysia/Indonesia), ranging from 2.8 to 10.6 ppbv decade$^{-1}$ for

IAGOS and 3.8 to 6.7 ppbv decade$^{-1}$ for ozonesondes in this period. In Europe and North America, however, we find much weaker tropospheric ozone trends and some inconsistency in the sign of tropospheric ozone trends derived from IAGOS and ozonesondes. Six ozonesonde stations in the Southern Hemisphere have increasing trends in free tropospheric ozone, with the strongest trends at the low latitude sites of La Réunion and Natal. No significant tropospheric ozone increases are found at high-latitudes in both hemispheres in 1995–2017.

570 Our GEOS-Chem simulation driven by reanalysis meteorological fields and the most up-to-date year-specific anthropogenic emission inventory reproduces the large tropospheric ozone increases over the tropics in 1995–2017. GEOS-Chem simulated trends in tropospheric ozone account for 51.8-81.4% of the IAGOS trends over East Asia, India, Southeast Asia, Persian Gulf, and Malaysia/Indonesia, and also catches the positive tropospheric ozone trends at three ozonesonde stations in East Asia, but trends are largely underestimated. Comparisons of observed vs modelled ozone values in 1995–1999 suggest that emissions

575 in the early years in developing regions are likely overestimated and contribute to the underestimation of tropospheric ozone trends. In the northern middle and high latitudes, the model shows no notable tropospheric ozone trends. GEOS-Chem estimates an increasing trend in global tropospheric ozone burden of 0.2 Tg year$^{-1}$ in 1995–2014 (0.4 Tg year$^{-1}$ in 1995–2017), compared to the CMIP6 model ensemble of 0.4 to 1.3 Tg year$^{-1}$ in 1995–2014. The smaller tropospheric ozone trends in GEOS-Chem compared to most of the CMIP6 models are partly due to the smaller trends in anthropogenic emissions of ozone

580 precursors in the CEDSv2 inventory than the CEDS$_{CMIP6}$ inventory used in the CMIP6 models, and also because GEOS-Chem better captures the observed ozone decreases in the lower stratosphere.

We find that increases in the global anthropogenic emissions, including surface emissions of short-lived ozone precursors, aircraft emissions, and methane, contribute to increases in the tropospheric ozone burden by 0.5 Tg year$^{-1}$, acting as the dominant driver of the tropospheric ozone increase in 1995–2017. The larger emission-driven tropospheric ozone trends are

585 found in the developing regions in the low-latitudes, where emissions of ozone precursors can produce ozone at higher efficiency due to the higher solar radiation and NO$_x$ sensitivity compared to those at high-latitudes, and can effectively influence the global ozone burden through deep convection. In particular, we find a previously underappreciated contribution of aircraft emissions to the tropospheric ozone increase (0.3 Tg year$^{-1}$), accounting for 66% of the total emission-driven tropospheric ozone trends. This large contribution is because aircraft NO$_x$ emissions are mainly released in the mid- and upper

590 troposphere, where water vapor content is lower, the NO$_x$ level is low, and lifetime of NO$_x$ is longer, leading to higher ozone production efficiency. Climatic and stratospheric factors contribute to a reduction of tropospheric ozone over mid-latitudes and high latitudes of both hemispheres in the absence of anthropogenic emission changes in 1995–2017. Ozone decreases in the lower stratosphere and a negative trend in STE flux in 1995–2017 may explain this decrease in ozone at mid- and high-latitudes. Climate variability such as ENSO largely influences the variability of tropospheric ozone through modulating

595 biomass burning emissions.

We also examine the radiative impacts of tropospheric ozone changes in 1995–2017. We estimate a global mean tropospheric ozone total radiative impact of 18.5 mW m$^{-2}$ in 2013–2017 compared to 1995–1999 level, with an increase by ~1.2%, but we suggest the true radiative impacts should be larger as our simulation underestimates the overall tropospheric ozone trends from 1995-2017. Changes in anthropogenic emissions are the dominant factor driving the increase in ozone radiative impact from

600 1995. The increase is mainly attributed to the equatorward redistribution of emissions to areas with strong convection and the

increase in aircraft emissions; both contribute to the increase in ozone in the mid- and upper troposphere and over the tropics where the potential ozone radiative impacts are a magnitude of two larger than those at the surface over mid-latitudes.

Our study thus highlights the dominant contribution of changes in global anthropogenic emission patterns, including the equatorward redistribution of surface emissions and the rapid increases in aircraft emissions, to the increases in tropospheric ozone and resulting radiative impacts in 1995–2017. Uncertainties in the anthropogenic emission inventory, especially for the early period and for developing regions where activity data are less effectively collected/constrained, may lead to the underestimation of GEOS-Chem simulated tropospheric ozone trends, especially in the tropics. Using long-term satellite observations of $NO_x$ as a top-down constraint on trends in anthropogenic emissions (Qu et al., 2020; Chen et al., 2021) may help to improve the model's ability to capture observed ozone trends. The spatial resolution of $4° \times 5°$ in our simulations limits the model ability to capture finer-scale ozone trends. We also call for more modeling studies to better understand ozone variability in the lower stratosphere and to quantify its impact on tropospheric ozone trends.

**Data availability.** The ozonesonde data are from https://woudc.org/data.php. The IAGOS data set is archived at http://www.iagos-data.fr/. The updated global anthropogenic emissions data from CEDS are available from https://doi.org/10.25584/PNNLDataHub/1779095. The MERRA-2 reanalysis data are from http://geoschemdata.computecanada.ca/ExtData/GEOS_4x5/MERRA2/. The CMIP6 model outputs are available on the Earth System Grid Federation (ESGF) website (https://esgf-data.dkrz.de/search/cmip6-dkrz/). The tropopause pressure data from MERRA-2 are available at https://doi.org/10.5067/AP1B0BA5PD2K. Data from GEOS-Chem modelling that support the findings of this study can be accessed by contacting the corresponding authors (Xiao Lu, luxiao25@mail.sysu.edu.cn; Shaojia Fan, eesfsj@mail.sysu.edu.cn).

**Author contributions.** XL, SJF, and HLW designed the study. HLW conducted the modeling and data analyses with contributions from BSS and KW. DJJ, ORC, KLC, KL, MG, and YML advised the trend analyses and model interpretation. TWW and JZ contributed to the CMIP6 model result and its interpretation. BS, PN, and RB contributed the IAGOS data and advised its interpretation. HLW, XL, and SJF wrote the paper with input from all authors. All authors contributed to the discussion and improvement of the paper.

**Competing interests.** The authors declare that they have no conflict of interest.

**Acknowledgments.** The authors acknowledge the strong support of the European Commission, Airbus and the airlines (Lufthansa, Air France, Austrian, Air Namibia, Cathay Pacific, Iberia and China Airlines, so far) who have carried the

MOZAIC or IAGOS equipment and performed the maintenance since 1994. In its last 10 years of operation, MOZAIC has been funded by INSU-CNRS (France), Météo-France, Université Paul Sabatier (Toulouse, France) and the Jülich Research Center (FZJ, Jülich, Germany). IAGOS has been additionally funded by the EU projects IAGOS-DS and IAGOS-ERI. The MOZAIC-IAGOS database is supported by AERIS, the French portal for data and service for the atmosphere. The data are available at https://www.iagos.org/ thanks to additional support from AERIS. O.R. Cooper and K.-L. Chang were supported by the NOAA Cooperative Agreement with CIRES, NA17OAR4320101. The authors thank David Lee, Ruijun Dang, Sebastian Eastham, and Viral Shah for valuable discussions on aircraft emissions.

**Financial support.** This research has been supported by the National Natural Science Foundation of China (NSFC, grant no. 41030164), the Key-Area Research and Development Program of Guangdong Province (grant no. 2020B1111360003), Guangdong Major Project of Basic and Applied Basic Research (grant no. 2020B0301030004), the Guangdong science and technology plan project (grant no. 2019B121201002), and the Fundamental Research Funds for the Central Universities, Sun Yat-sen University (grant no. 22qntd1908).

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

 **Table 1: List of the monitoring ozonesonde stations.**

| No. | Site | Region | Latitude (°) | Longitude (°) | Elevation (m) | Sample frequency (per month) |
|---|---|---|---|---|---|---|
| 1 | Alert | Canada | 82.5 | -62.3 | 210 | 4.0 |
| 2 | Eureka | Canada | 80.05 | -86.42 | 610 | 5.6 |
| 3 | Resolute | Canada | 74.72 | -94.98 | 64 | 3.0 |
| 4 | Edmonton | Canada | 53.55 | -114.10 | 766 | 4.0 |
| 5 | Goose Bay | Canada | 53.30 | -60.39 | 39 | 3.9 |
| 6 | Boulder ESRL HQ (CO) | United States | 39.99 | -105.26 | 1634 | 4.5 |
| 7 | Legionowo | Poland | 52.40 | 20.97 | 96 | 4.5 |
| 8 | De Bilt | Netherlands | 52.10 | 5.18 | 2 | 4.4 |
| 9 | Uccle | Belgium | 50.80 | 4.36 | 100 | 12.2 |
| 10 | Praha | Czech Republic | 50.01 | 14.45 | 302 | 4.0 |
| 11 | Hohenpeissenberg | Germany | 47.80 | 11.01 | 985 | 10.6 |
| 12 | Payerne | Switzerland | 46.81 | 6.94 | 490 | 12.8 |
| 13 | Madrid | Spain | 40.45 | -3.72 | 680 | 3.7 |
| 14 | Sapporo | Japan | 43.06 | 141.33 | 26 | 3.7 |
| 15 | Tateno (Tsukuba) | Japan | 36.10 | 140.13 | 31 | 4.3 |
| 16 | Naha | Japan | 26.20 | 127.68 | 27 | 3.5 |
| 17 | Hilo (HI) | Northeast Pacific | 19.58 | -155.07 | 11 | 5.8 |
| 18 | Paramaribo | Suriname | 5.81 | -55.21 | 23 | 6.2 |
| 19 | Nairobi | Kenya | -1.30 | 36.75 | 1795 | 3.8 |
| 20 | Natal | Brazil | -6.00 | -35.20 | 0 | 3.1 |
| 21 | Samoa (Cape Matatula) | Southeast Pacific | -14.25 | -170.56 | 77 | 4.4 |

| 22 | La Réunion | Southwest Indian Ocean | -21.08 | 55.38 | 2160 | 3.1 |
| 23 | Broadmeadows | Australia | -37.69 | 144.95 | 108 | 3.8 |
| 24 | Macquarie Island | Australia | -54.50 | 158.94 | 6 | 3.7 |
| 25 | Lauder | New Zealand | -45.04 | 169.68 | 370 | 4.3 |
| 26 | Marambio | Antarctica | -64.24 | -56.62 | 198 | 4.6 |
| 27 | Syowa | Antarctica | -69.00 | 39.58 | 21 | 4.6 |

**Table 2: Configurations of GEOS-Chem simulations in this study[a].**

| Simulation | Aircraft emissions | Anthropogenic emissions | Biomass burning emissions | Global methane concentrations | Meteorology |
|---|---|---|---|---|---|
| BASE | V | V | V | V | V |
| FixAircraft | 1995 | V | V | V | V |
| FixAC | 1995 | 1995 | V | 1995 | V |
| FixABC | 1995 | 1995 | 1995 | 1995 | V |

[a] 'V' denotes those specific inputs vary interannually in the simulation, and '1995' denotes that the inputs are fixed to 1995 conditions.

[b] 'FixAircraft' denotes only global aircraft emissions are fixed at 1995 levels in the simulation. 'FixAC' denotes global anthropogenic emissions (including aircraft emissions) and methane concentration level are fixed to 1995 conditions. 'FixABC' denotes biomass burning emissions are fixed at 1995 levels based on FixAC. Ozone trends from aircraft emissions

**Table 3: Information on the CMIP6 models used in this study.**

| No. | CMIP6 Model | Ensemble member [a] | longitude × latitude | Vertical levels (top level) | Reference |
|-----|-------------|---------------------|----------------------|-----------------------------|-----------|
| 1 | BCC-ESM1 | r1i1p1f1 | ~2.8° × 2.8° | L26 (2.19 hPa) | Wu et al. (2020) |
| 2 | CESM2 | r1i1p1f1 | ~1.25° × 0.9° | L32 (2.25 hPa) | Danabasoglu (2019a) |
| 3 | CESM2-WACCM | r1i1p1f1 | ~1.25° × 0.9° | L70 ($4.5 \times 10^{-6}$ hPa) | Danabasoglu (2019b) |
| 4 | GFDL-ESM4 | r1i1p1f1 | 1.25° × 1° | L49 (0.01 hPa) | Krasting et al. (2018) |
| 5 | IPSL-CM6A-LR | r1i1p1f1 | ~2.5° × 1.26° | L79 (0.01 hPa) | Boucher et al. (2018) |
| 6 | MPI-ESM-1-2-HAM | r1i1p1f1 | ~1.8° × 1.8° | L47 (0.01 hPa) | Neubauer et al. (2019) |
| 7 | NoRESm2-MM | r1i1p1f1 | ~1.25° × 0.9° | L32 (0.03 hPa) | Bentsen et al. (2019) |

[a] There are 4 indices defining an ensemble member: "r" for realization, "i" for initialization, "p" for physics, and "f" for forcing.


**Table 4: Annual trends and 2-sigma uncertainty (ppbv decade$^{-1}$) in median ozone for tropospheric column (950-250 hPa) in 1995-2017 from both observations and GEOS-Chem.**

| Region | Measurements | Ozonesonde site or IAGOS region | Observation | | GEOS-Chem | |
|---|---|---|---|---|---|---|
| | | | Trend ± 2σ | *p*-value | Trend ± 2σ | *p*-value |
| East Asia | Ozonesonde | Sapporo | 3.73 ± 0.69 | < 0.01 | 1.82 ± 0.50 | < 0.01 |
| | Ozonesonde | Tateno (Tsukuba) | 4.75 ± 0.43 | < 0.01 | 1.36 ± 0.40 | < 0.01 |
| | Ozonesonde | Naha | 5.18 ± 0.40 | < 0.01 | 0.47 ± 0.41 | < 0.05 |
| | IAGOS | East Asia | 2.55 ± 0.15 | < 0.01 | 2.08 ± 0.17 | < 0.01 |
| India | IAGOS | India | 5.01 ± 0.43 | < 0.01 | 2.66 ± 0.36 | < 0.01 |
| Southeast Asia | IAGOS | Southeast Asia | 5.53 ± 0.26 | < 0.01 | 2.87 ± 0.23 | < 0.01 |
| Persian Gulf | IAGOS | Persian Gulf | 3.66 ± 0.26 | < 0.01 | 2.47 ± 0.19 | < 0.01 |
| Malaysia/Indonesia | IAGOS | Malaysia/Indonesia | 4.36 ± 0.41 | < 0.01 | 2.69 ± 0.33 | < 0.01 |
| Africa | IAGOS | Gulf of Guinea | 2.61 ± 0.34 | < 0.01 | 0.60 ± 0.27 | < 0.01 |
| | Ozonesonde | Nairobi | 1.66 ± 0.56 | < 0.01 | 0.14 ± 0.51 | 0.59 |
| South America | Ozonesonde | Paramaribo | 0.69 ± 0.63 | < 0.05 | 0.84 ± 0.55 | < 0.01 |
| | Ozonesonde | Natal | 3.00 ± 0.74 | < 0.01 | 0.89 ± 0.75 | < 0.05 |
| | IAGOS | Northern South America | 3.72 ± 0.50 | < 0.01 | 2.14 ± 0.56 | < 0.01 |
| Pacific | Ozonesonde | Hilo (HI) | 1.41 ± 0.46 | < 0.01 | 0.98 ± 0.46 | < 0.01 |
| | Ozonesonde | Samoa (Cape Matatula) | 0.83 ± 0.36 | 0.25 | -0.60 ± 0.38 | < 0.01 |
| Europe | Ozonesonde | Legionowo | -0.50 ± 0.42 | < 0.05 | 0.62 ± 0.35 | < 0.01 |
| | Ozonesonde | De Bilt | 2.14 ± 0.37 | < 0.01 | -0.15 ± 0.34 | 0.38 |
| | Ozonesonde | Uccle | 1.70 ± 0.23 | < 0.01 | 0.10 ± 0.20 | 0.34 |
| | Ozonesonde | Praha | 0.35 ± 0.43 | 0.11 | -0.22 ± 0.46 | 0.33 |
| | Ozonesonde | Hohenpeissenberg | 0.27 ± 0.23 | < 0.05 | -0.24 ± 0.21 | < 0.05 |

| | | | | | | |
|---|---|---|---|---|---|---|
| | Ozonesonde | Payerne | -0.63 ± 0.23 | < 0.01 | 0.36 ± 0.19 | < 0.01 |
| | Ozonesonde | Madrid | 0.38 ± 0.39 | < 0.05 | 0.48 ± 0.36 | < 0.01 |
| | IAGOS | Europe | 0.83 ± 0.06 | < 0.01 | -0.41 ± 0.05 | < 0.01 |
| United States | Ozonesonde | Boulder ESRL HQ (CO) | -0.64 ± 0.34 | < 0.01 | 0.009 ± 0.35 | 0.96 |
| | IAGOS | Eastern North America | 0.96 ± 0.13 | < 0.01 | -0.70 ± 0.12 | < 0.01 |
| | IAGOS | Southeast US | 0.86 ± 0.22 | < 0.01 | -1.02 ± 0.16 | < 0.01 |
| | IAGOS | Western North America | 1.67 ± 0.32 | < 0.01 | -2.03 ± 0.37 | < 0.01 |
| Canada | Ozonesonde | Alert | -0.07 ± 1.22 | 0.91 | 0.11 ± 1.34 | 0.87 |
| | Ozonesonde | Eureka | 0.35 ± 1.13 | 0.53 | 0.39 ± 1.22 | 0.52 |
| | Ozonesonde | Resolute | -0.65 ± 1.91 | 0.49 | 1.56 ± 1.37 | < 0.05 |
| | Ozonesonde | Edmonton | 1.80 ± 0.57 | < 0.01 | 0.66 ± 0.61 | < 0.05 |
| | Ozonesonde | Goose Bay | 2.22 ± 0.74 | < 0.01 | -0.89 ± 0.63 | < 0.01 |
| Southern Hemisphere | Ozonesonde | La Réunion | 4.60 ± 0.93 | < 0.01 | 0.94 ± 0.95 | < 0.05 |
| | Ozonesonde | Broadmeadows | -0.36 ± 0.52 | 0.17 | -0.96 ± 0.72 | < 0.01 |
| | Ozonesonde | Macquarie Island | -0.85 ± 0.43 | < 0.01 | -1.14 ± 0.76 | < 0.01 |
| | Ozonesonde | Lauder | 0.34 ± 0.30 | < 0.05 | 0.43 ± 0.51 | 0.09 |
| | Ozonesonde | Marambio | -0.43 ± 0.36 | < 0.05 | 0.17 ± 0.71 | 0.62 |
| | Ozonesonde | Syowa | 0.003 ± 0.37 | 0.99 | -0.33 ± 0.54 | 0.22 |

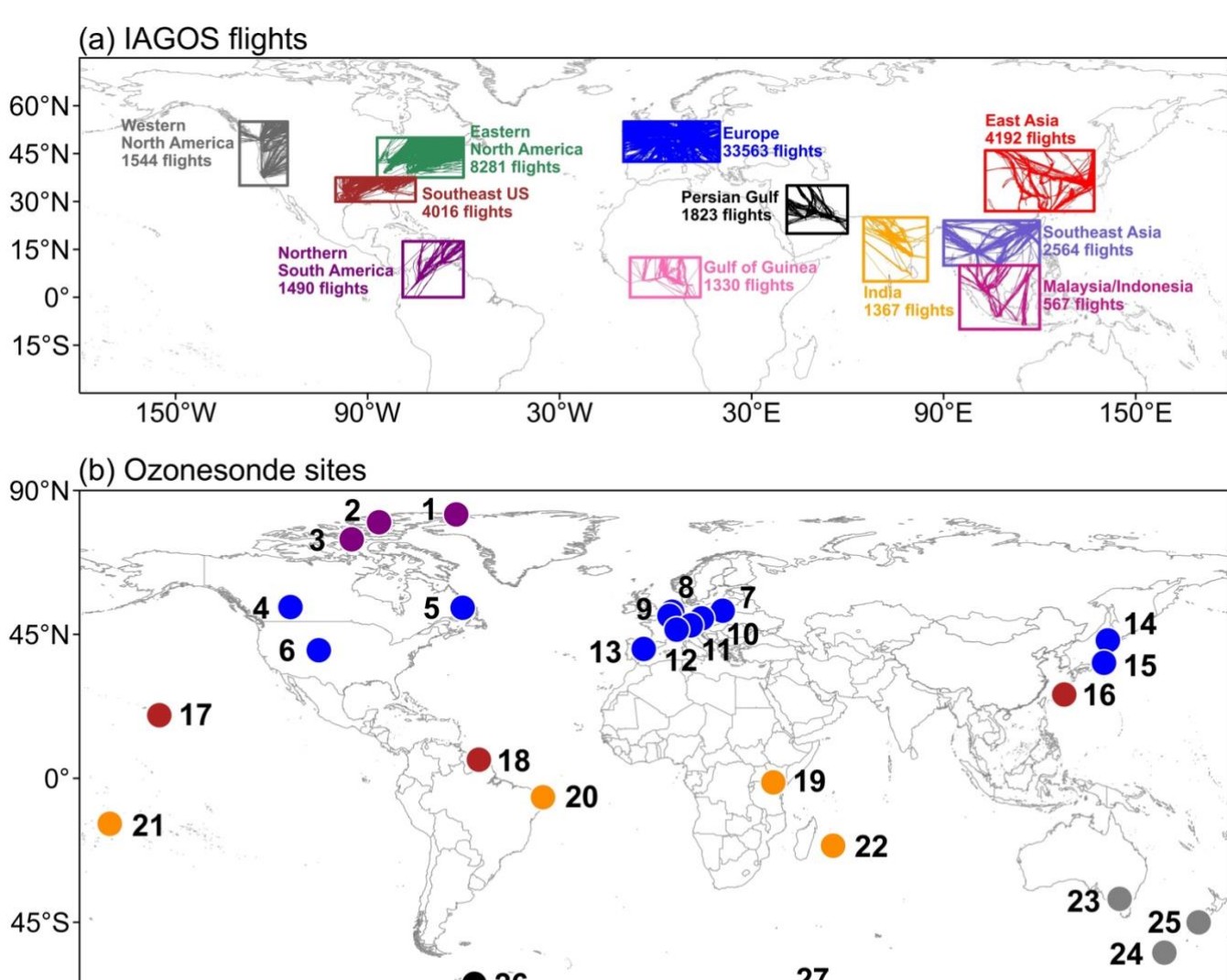

**Figure 1: IAGOS and ozonesonde measurements of tropospheric ozone used in this study. The upper panel shows the map of the 11 study regions with frequent IAGOS sampling between 1995 and 2017 (grouped by color). The flight tracks are indicated in the boxes showing western North America, eastern North America, Europe, East Asia (including the North China, Korea, and part of Japan), Southeast United States, northern South America, Gulf of Guinea, the Persian Gulf, India, Southeast Asia, and Malaysia/Indonesia. The lower panel shows the location of selected ozonesonde sites in 1995–2017 used in this study, grouped by six latitude bands with**
**an interval of 30° as denoted by different colors.**

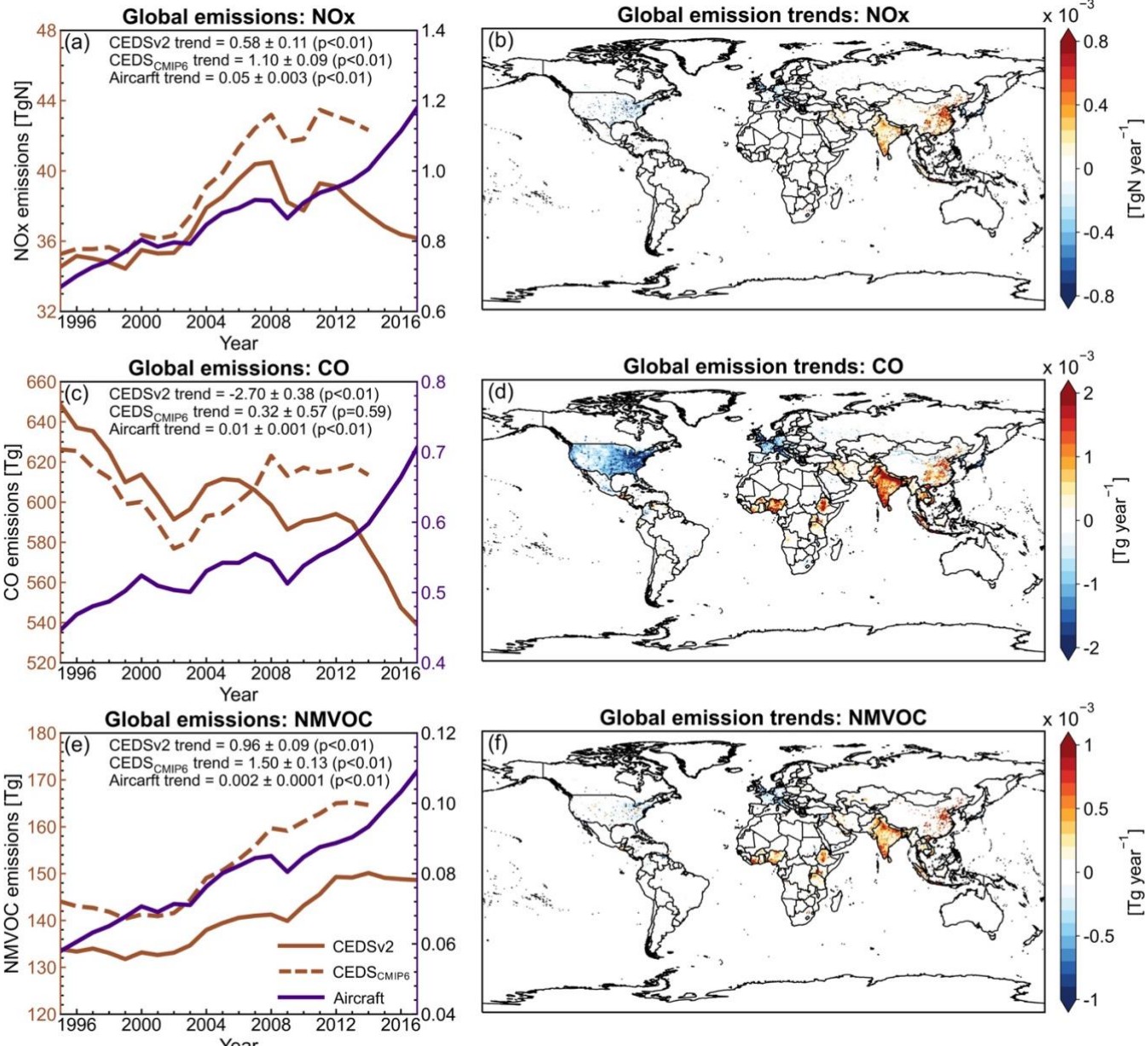

**Figure 2: Trends in global annual anthropogenic (excluding aircraft emissions) and aircraft emissions of NOₓ, CO, and NMVOCs from 1995 to 2017. The left panels show the total global anthropogenic NOₓ, CO and NMVOC emissions from the CEDSv2 and CEDS_CMIP6 inventories, and the right panels show the spatial distribution of emission trends in the CEDSv2 inventory. Aircraft emissions are from O'Rourke et al. (2021). The total global anthropogenic emission trends with *p*-value are shown in left panels.**


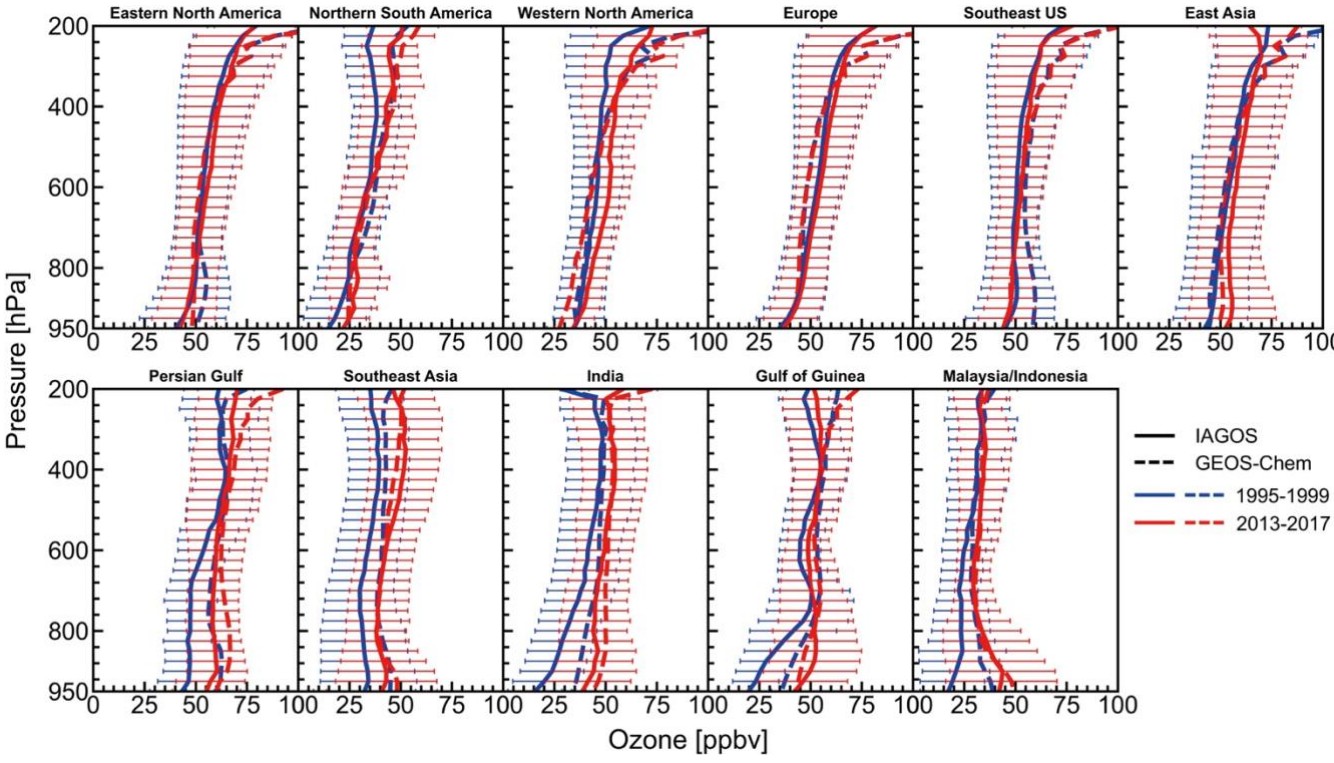

**Figure 3: Comparison of IAGOS observations (solid line) and simulated (dashed line) ozone vertical profiles for 11 IAGOS regions from 1995-1999 (blue line) to 2013-2017 (red line). Horizontal bars are standard deviations in the observations.**

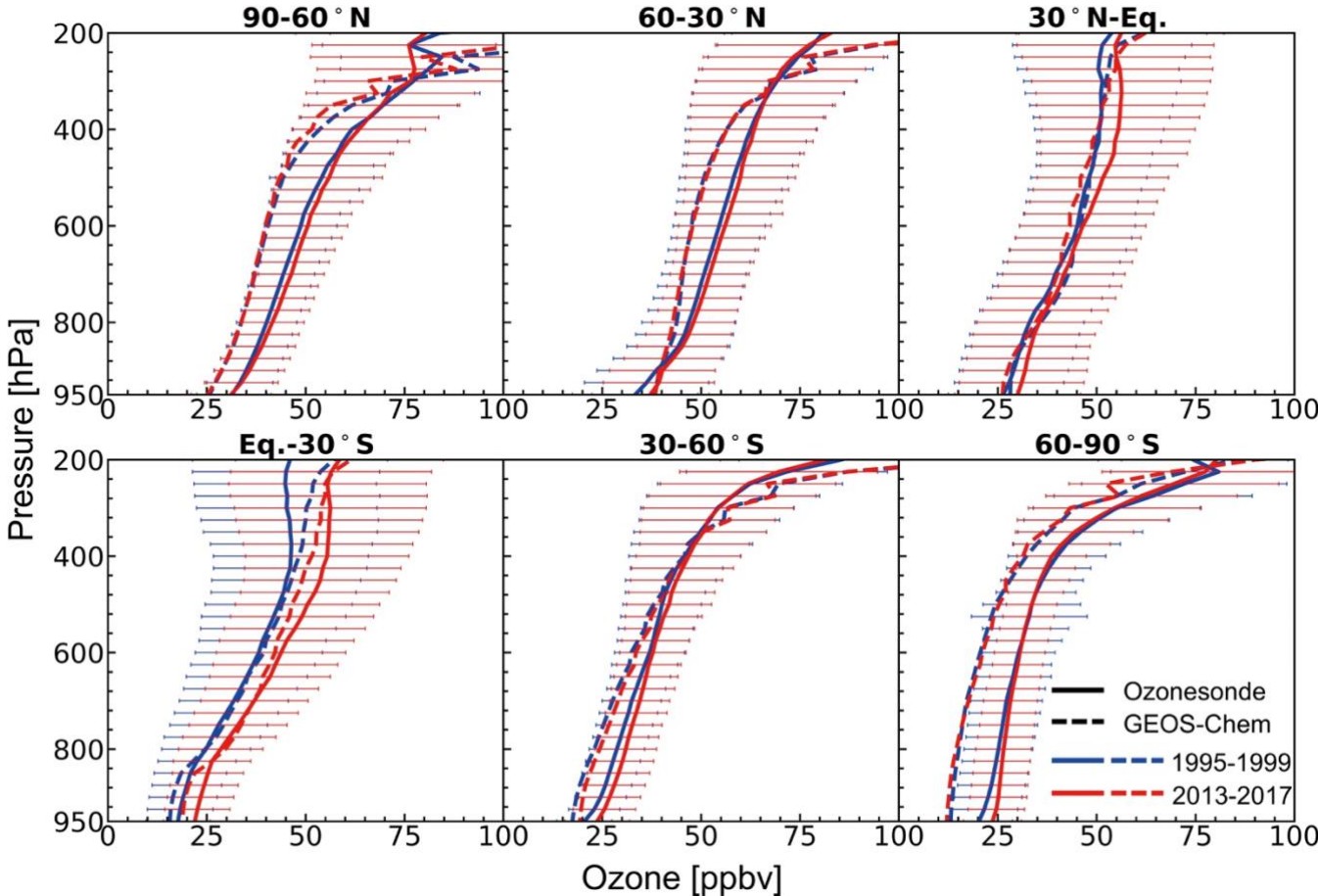

**Figure 4: Comparison of ozonesonde (solid line) and simulated (dashed line) ozone vertical profiles for six zonal bands over 1995-1999 (blue line) and 2013-2017 (red line). Horizontal bars are standard deviations in the observation.**

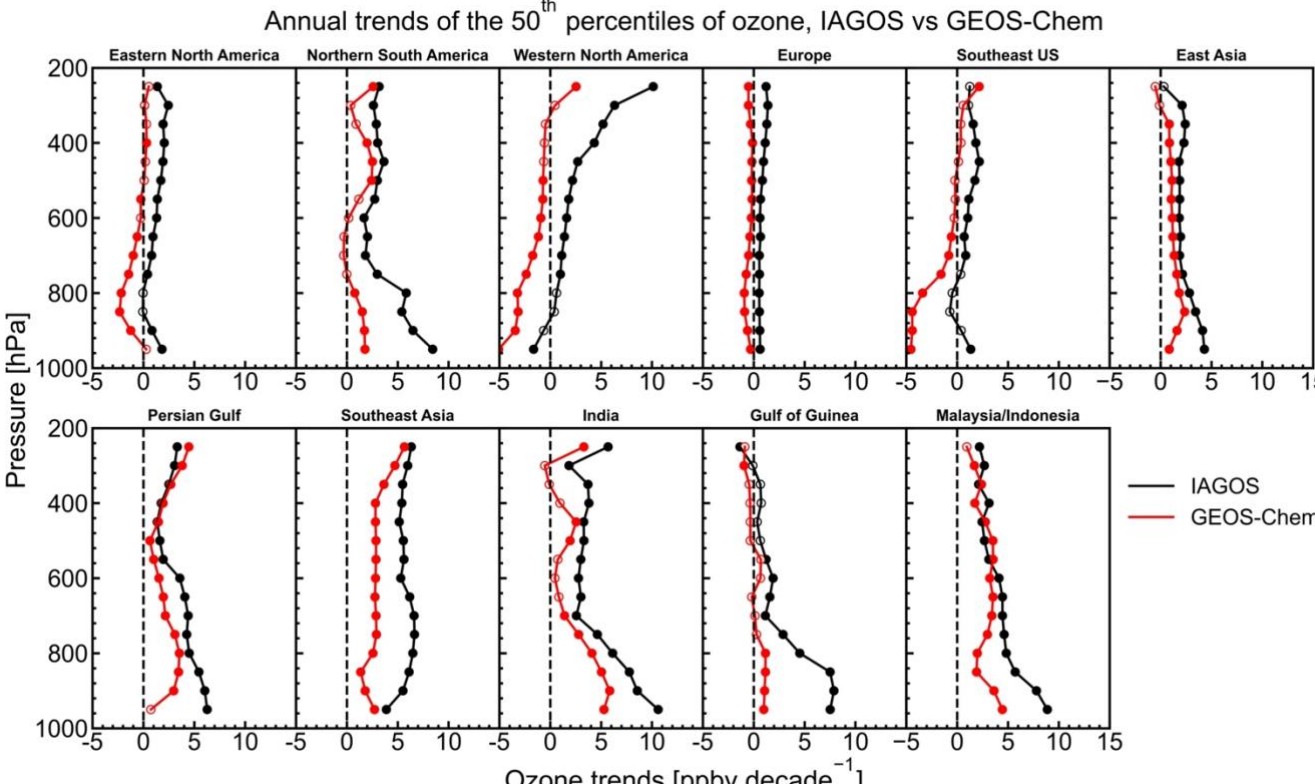

**Figure 5: Annual trends of the 50th percentiles of IAGOS observed and GEOS-Chem simulated ozone (ppbv decade⁻¹) at intervals of 50 hPa. The trends are calculated between 1995 and 2017 above the 11 selected regions (Fig.1) using the quantile regression method (Section 2.5). Filled circles indicate trends with *p*-value less than 0.05.**

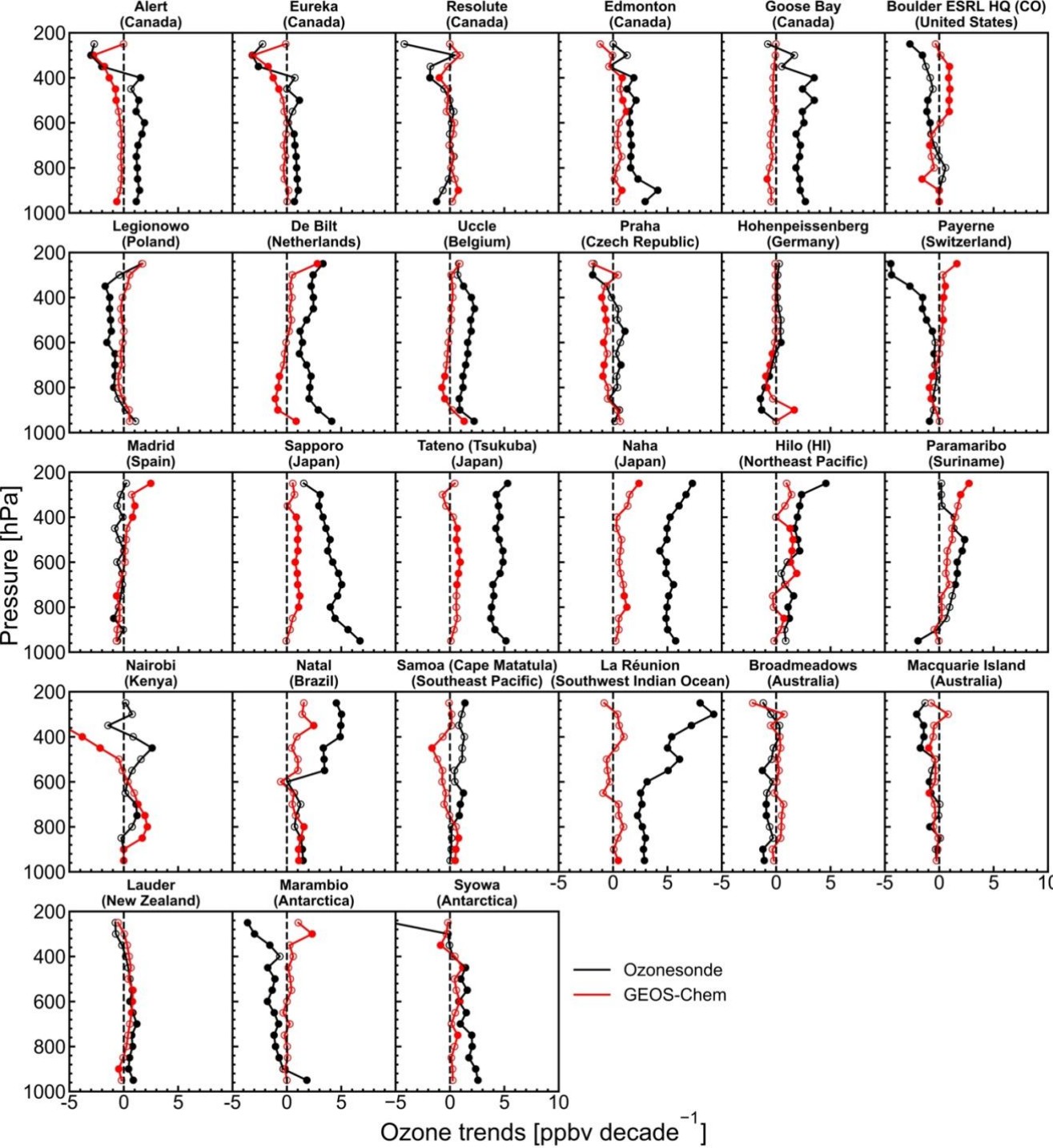

**Figure 6: Same as Figure 5 but for the comparison of ozonesonde versus GEOS-Chem results.**

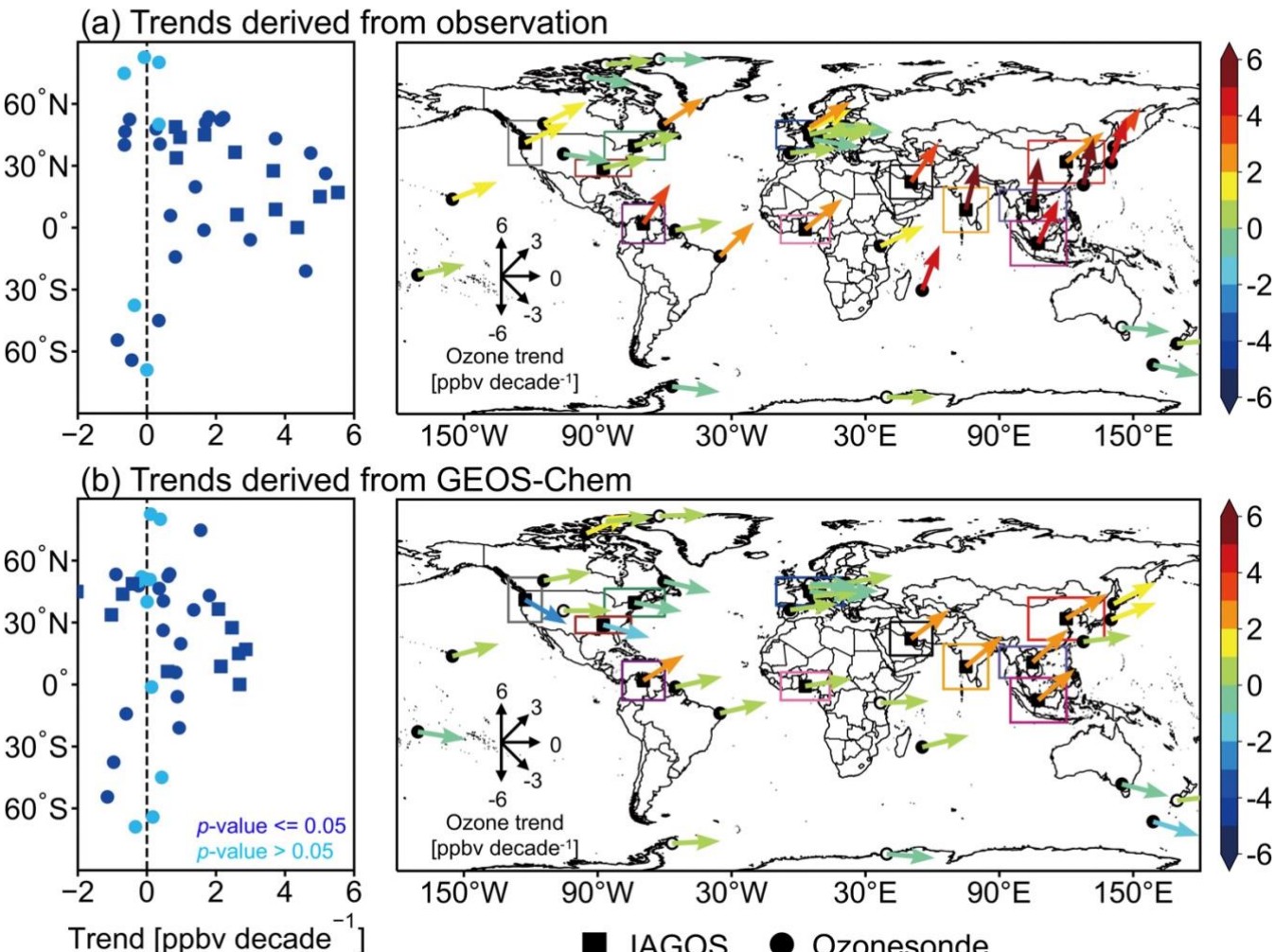

**Figure 7:** Comparison of tropospheric ozone trends of the 50[th] percentiles of tropospheric column (950 to 250 hPa) in 1995-2017 derived from observation (top panel) and GEOS-Chem (bottom panel). Trends are estimated for 11 selected IAGOS regions and ozonesonde sites with frequent sampling. Symbol colors indicate the p-value associated with the trend at each site and region in the left panel. Both directions and colors of the vectors in the right panel indicate the ozone change rates in ppbv decade[-1]. Dark colors (left panel) and filled circles or squares (right panel) indicated trends with *p*-value < 0.05.

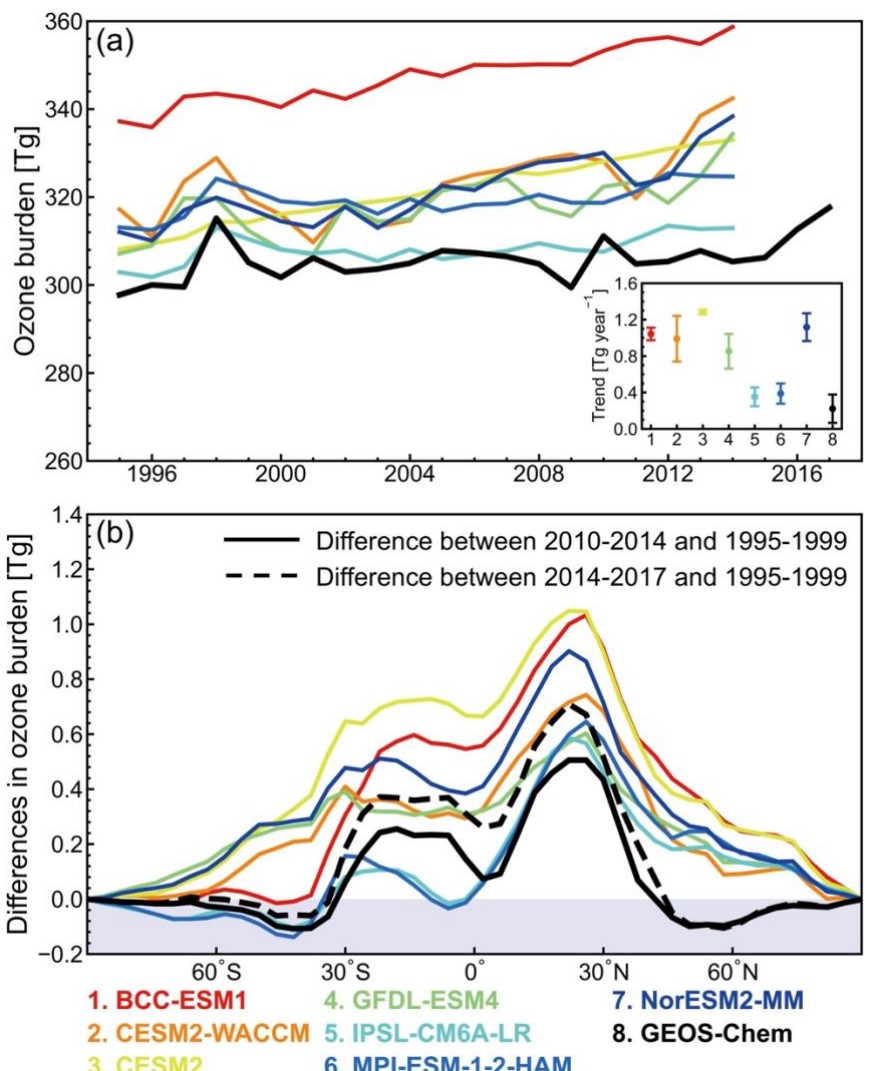

Figure 8: Evolution of 1995-2017 tropospheric ozone burden from GEOS-Chem and 7 CMIP6 models (available for 1995-2014) used in this study. Panel (a) shows the time series of tropospheric ozone burden integrated from 90°S to 90°N for the period 1995 to 2017. The black line represents the results of the GEOS-Chem simulation, and colored lines are from CMIP6 models. Dot plots show the tropospheric ozone burden trends for 1995-2014 in different models with the vertical bars showing the 95% confidence interval. Panel (b) shows the differences in zonal integrated tropospheric ozone burden for the nine models between 2010-2014 and 1995-1999 (solid line) and for GEOS-Chem between 2014-2017 and 1995-1999 (dashed line).

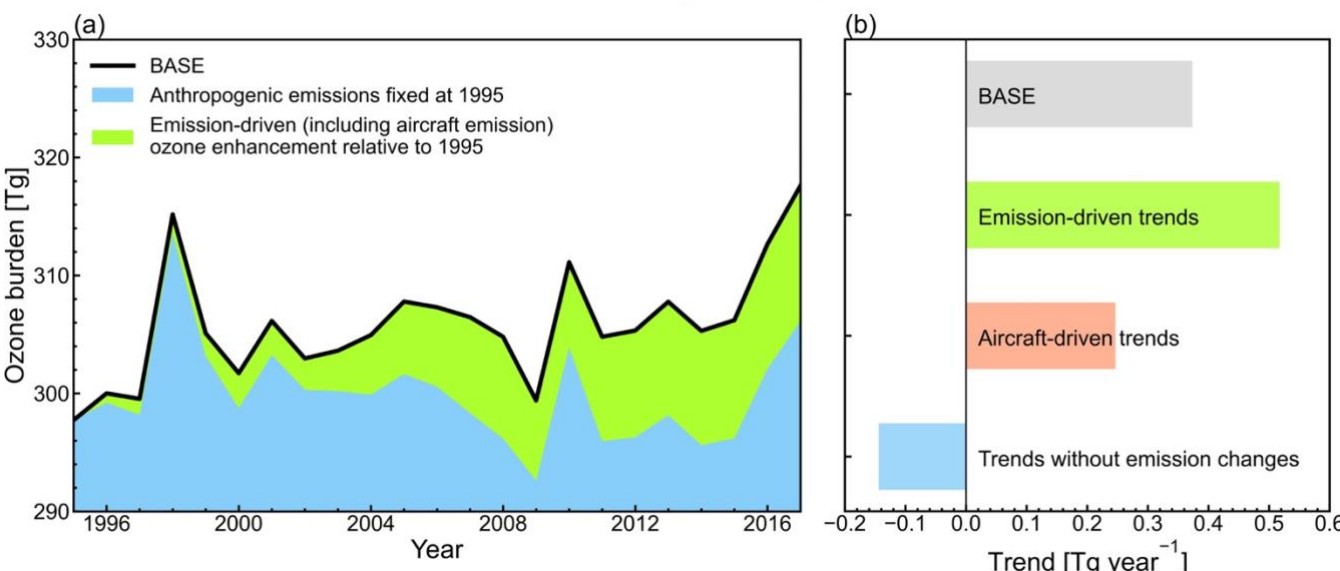

**Figure 9: Drivers of global tropospheric ozone burden and trends from 1995 to 2017 estimated in GEOS-Chem model. (a) Evolution of the GEOS-Chem simulated annual global tropospheric ozone burden (black line, same as Fig.8a) in the BASE simulation. The blue shadings show the evolution of tropospheric ozone burden from the FixAC simulation, estimating ozone burden if** 1190 **anthropogenic emissions (surface emissions, aircraft emissions, and methane) are fixed at the level of 1995. The green shadings thus estimate the tropospheric ozone burden contributed by the anthropogenic emission (including aircraft emission) changes relative to the 1995 level. Panel (b) shows the estimated tropospheric ozone trends from the BASE simulation and from different drivers.**

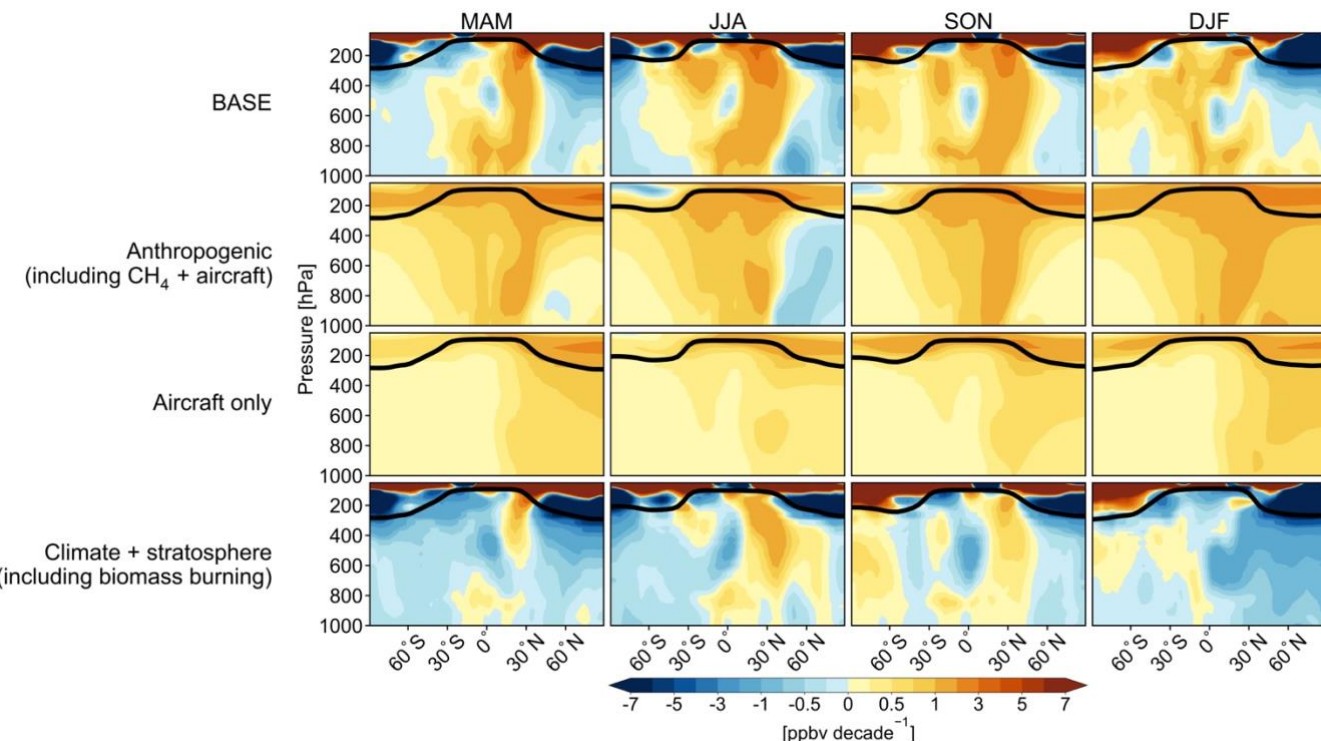

Figure 10: Drivers of seasonal zonal mean ozone trends from 1995 to 2017 estimated in GEOS-Chem model. Trends contributed by changes in anthropogenic emissions (surface emissions, aircraft emissions, and methane), aircraft emissions alone, and climatic and stratospheric factors are estimated. Black lines represent the 1995–2017 climatological seasonal mean tropopause from MERRA2 reanalysis dataset.

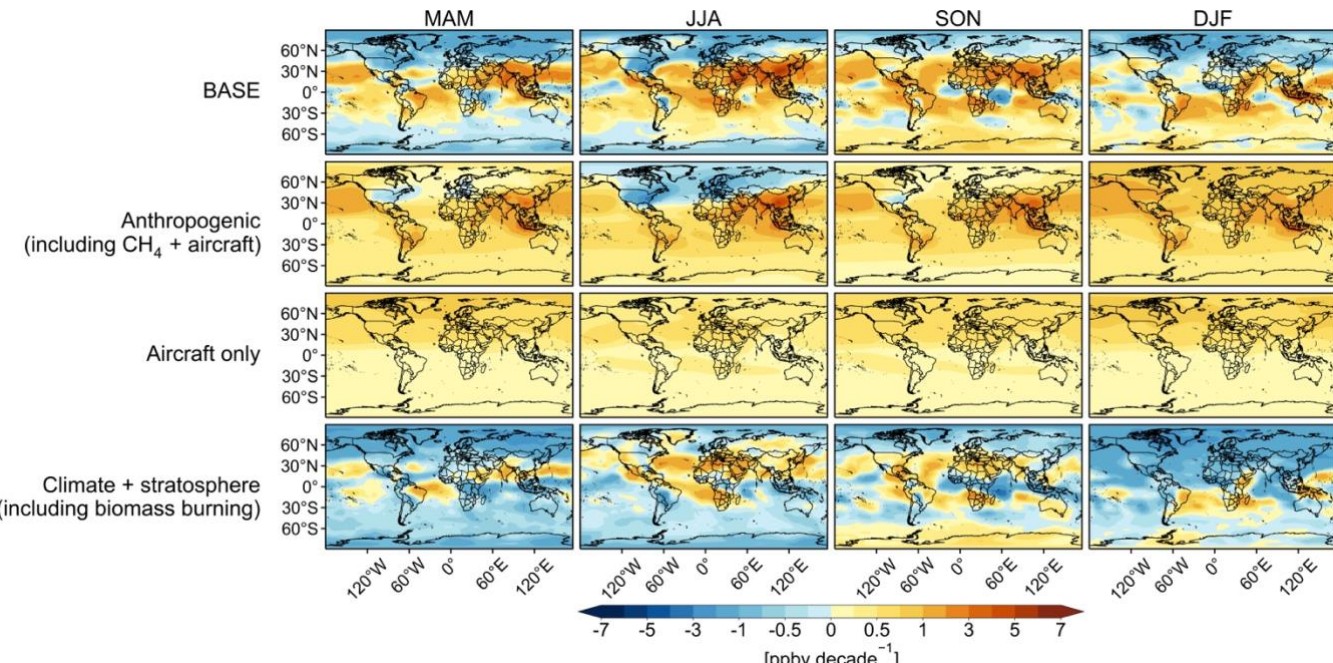

Figure 11: Drivers of seasonal mean tropospheric (950-250hPa) ozone trends from 1995 to 2017 estimated in GEOS-Chem model. Trends contributed by changes in anthropogenic emissions (surface emissions, aircraft emissions, and methane), aircraft emissions alone, and climatic and stratospheric factors are estimated.

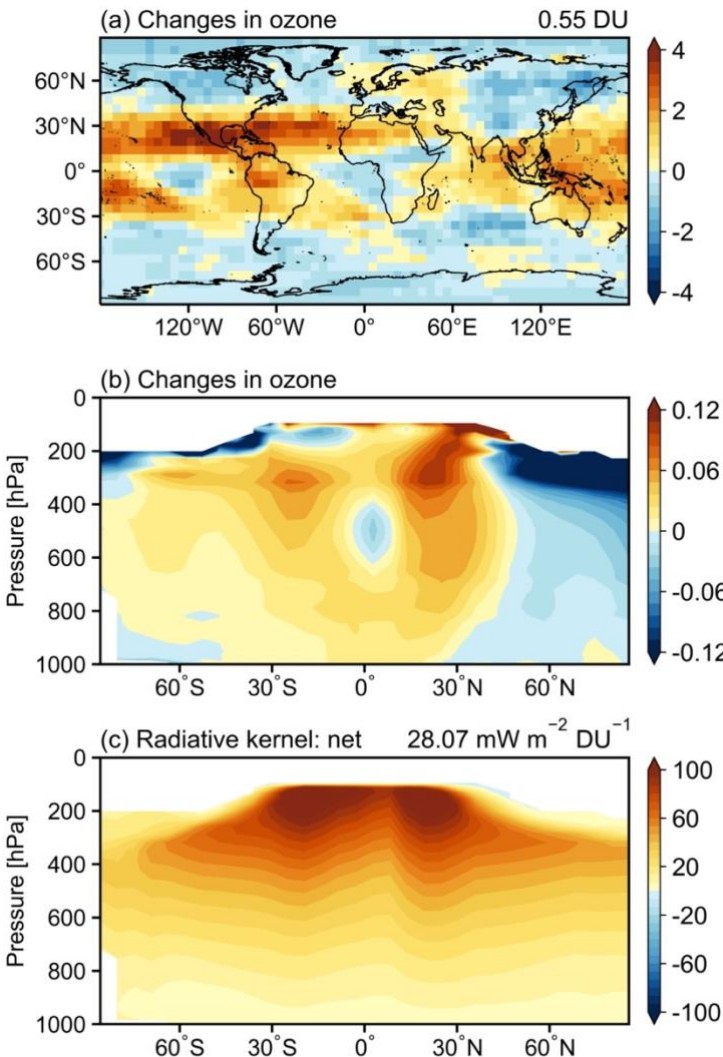

**Figure 12: Changes in annual mean tropospheric column ozone (a) and zonal mean ozone (b) between 2013-2017 and 1995-1999.** Panel (c) shows the values of the ozone radiative kernel (mW m$^{-2}$ DU$^{-1}$) for net forcing (LW + SW) from Skeie et al. (2020). The annual global mean values are shown in the upper right.

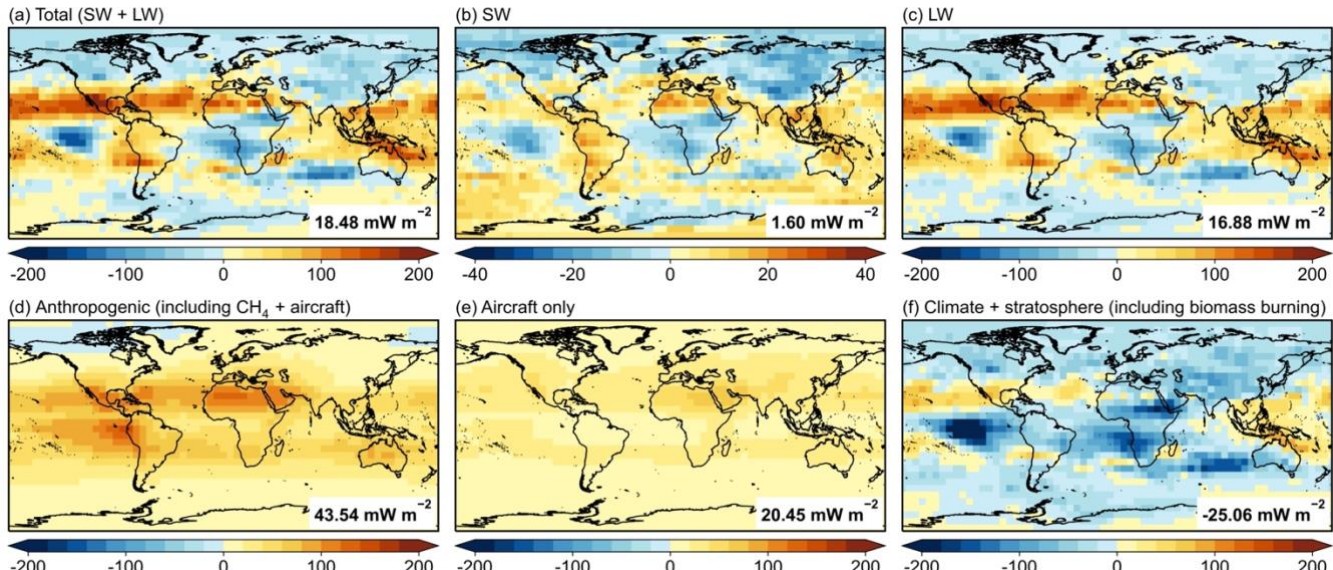

**Figure 13: Tropospheric ozone radiative impacts and attributions, 2013-2017 versus 1995-1999. Panel (a)-(c) shows the tropospheric ozone radiative impacts (mW m⁻²) for total (SW + LW) and for SW and LW, respectively. Panels (d)-(f) attribute the tropospheric ozone radiative impacts due to changes in anthropogenic emissions (including surface emissions, aircraft emission, global methane levels), aircraft emissions alone, and climate (including stratosphere and biomass burning) between 2013-2017 and 1995-1999. The annual global mean values are shown inset.**