# Peer review of "Global tropospheric ozone trends, attributions, and radiative impacts in 1995–2017: an integrated analysis using aircraft (IAGOS) observations, ozonesonde, and multi-decadal chemical model simulations"

_Atmospheric Chemistry and Physics, 2022_

## Referee Comment (RC2)

Review comments:

This study applied an extensive dataset, including aircraft, ozonesonde and global CTMs to study the trends of global ozone changes in the troposphere. Furthermore, the authors performed sensitivity simulations from the CTM to analyze the attribution of global ozone burden changes. They found a consistent ozone burden changes from both the aircraft and ozonesonde observations from 1995 to 2017 which was also captured by the CTM with latest emission inventory. The study also compared the trends simulated by the ensemble CMIP6 models, and concluded that the higher ozone trends from the CMIP6 was potential caused by the overestimation of anthropogenic emissions used in the earlier versions of CEDS emission inventory. The manuscript is generally well-written and the presentation quality is very good as well. I have two concerns about the methods in which I want the authors to address before this manuscript accepted for publication.

1. As comparison with the multi CMIP6 models, we can see the GEOS-Chem model used here has a really coarse resolution. I wonder how this issue will affect the trend analysis especially for regions in China and India which are experiencing significant changes both in climate and emissions at urban cores.
2. When doing the attributing analysis, taking aircraft for example, the authors fixed all the other emissions at 1995 level, and then varying the aircraft emissions. Since the world is experiencing significant emission changes for both developed regions (emission decreasing) and developing regions (emission increasing), so how this method will affect the contribution from the aircraft for the ozone production without considering the realistic emissions in specific years?

**Editorial comments:**

L55-56: rephrase this sentence.

L57: I feel change to "The ozone lifetime ranges/spans from xxx to …" reads better. Just a suggestion.

L70-79: I would encourage the authors to summary the findings from the IPCC report, instead of citing the whole paragraph for their own paper.

L202: The units for the trend should be "Tg yr$^{-1}$" or "Tg decade$^{-1}$".

L214: change to "(Zheng et al., 2018)"
Fig. 1:  Explain the color indications for low panel

Fig. 2: Explain the meanings in the bracket in left panels. The unit for the right panel is not accurate. Also reading the right panels for the trends of NOx CO and NMVOC, the authors can use the unit of "Tg/decade".

---

## Author Comment (AC1)

**Dear Editor Dr. Leiming Zhang,**

**Thank you very much for handling our manuscript. Please find below our itemized responses to the reviewers' comments and a marked-up manuscript. We have addressed all the comments raised by the reviewer and incorporated them in the revised manuscript.**

**Thank you for your consideration.**

**Sincerely,**
**Haolin Wang et al.**
* * *
**Reviewer #1**

**Comment [1-1]:** This study applied an extensive dataset, including aircraft, ozonesonde and global CTMs to study the trends of global ozone changes in the troposphere. Furthermore, the authors performed sensitivity simulations from the CTM to analyze the attribution of global ozone burden changes. They found a consistent ozone burden changes from both the aircraft and ozonesonde observations from 1995 to 2017 which was also captured by the CTM with latest emission inventory. The study also compared the trends simulated by the ensemble CMIP6 models, and concluded that the higher ozone trends from the CMIP6 was potential caused by the overestimation of anthropogenic emissions used in the earlier versions of CEDS emission inventory. The manuscript is generally well-written and the presentation quality is very good as well. I have two concerns about the methods in which I want the authors to address before this manuscript accepted for publication.

**Response [1-1]: We thank the reviewer for the positive comments on our study. Below, we provide a point-by-point response to the reviewer's comments and summarize the changes that have been made in the revised manuscript.**

**Comment [1-2]:** As comparison with the multi CMIP6 models, we can see the GEOS-Chem model used here has a really coarse resolution. I wonder how this issue will affect the trend analysis especially for regions in China and India which are experiencing significant changes both in climate and emissions at urban cores.

**Response [1-2]: We thank the reviewer for bringing this up. While we can principally apply the GEOS-Chem model at a finer global $2°\times2.5°$ resolution, we choose the $4°\times 5°$ resolution to reduce simulation time. Our simulation includes a 10-year spin-up run and four sets of 23-year run (1995-2017). The simulations were conducted consecutively without any break to make sure the modelled ozone trends are consistent. It takes four hours with 48 CPUs for a one-month GEOS-Chem standard simulation at $4°\times 5°$ (http://wiki.seas.harvard.edu/geos-chem/index.php/GEOS-Chem_13.3.0#1-month_benchmarks), so a 33-year simulation costs 66 natural days to finish. Running the model at $2° \times 2.5°$ would cost at least eight times as much computational time and resources, which is rather inapplicable.**

**We agree with the reviewer that coarse-grid simulations ($4°\times 5°$) may limit the ability of the model to capture finer-scale ozone trends, in particular at near surface where ozone and its precursor has a short lifetime. Artificial mixing of surface ozone precursors in coarse model grids may lead to higher-than-actual ozone production efficiency and therefore positive ozone biases (Wild and Prather, 2006; Yu et al., 2016; Young et al, 2018), especially for grids covering urban cores in China and India where significant changes in emissions are occurring. In light of this we do not use surface observations for model evaluation, and have limited the discussion of surface ozone trends. The limitation of model resolution, however,**

should be alleviated for ozone in the free troposphere, where ozone has longer chemical lifetime so that we expect ozone there is better mixed than at near surface (Petetin et al., 2016). Nevertheless, increasing model spatial and temporal resolution is still preferable in future modeling studies of long-term ozone trends.

We have clarified in the text: **"We run the GEOS-Chem model at a horizontal resolution of 4° (latitude) × 5° (longitude), with 72 vertical layers extending from surface to 0.01 hPa. One-month model simulation at this resolution costs 4 hours with 48 CPUs (http://wiki.seas.harvard.edu/geos-chem/index.php/GEOS-Chem_13.3.0#1-month_benchmarks). Yielding 33-year (including 10-year spin-up simulation) global simulation of ozone trends thus require computation time of more than 60 natural days. As such we do not use a finer resolution of 2°× 2.5° that would otherwise cost at least eight times as much computational time and resources as in this study. This relatively coarse resolution of 4°× 5° may limit the ability of the model to capture finer-scale ozone trends, in particular at near surface where ozone and its precursor has a short lifetime. Artificial mixing of surface ozone precursors in coarse model grids may lead to higher-than-actual ozone production efficiency and therefore positive ozone biases which may further influence trend analyses (Wild and Prather, 2006; Yu et al., 2016; Young et al, 2018; Yin et al., 2021). The limitation of model resolution, however, should be alleviated for ozone in the free troposphere, where ozone has longer chemical lifetime and should be better mixed than at near surface (Petetin et al., 2016). In light of this we do not use surface ozone observations for model evaluation, and mainly focus the trend analyses on above 950 hPa."**

Reference:

Petetin, H., Thouret, V., Athier, G., Blot, R., Boulanger, D., Cousin, J. M., Gaudel, A., Nédélec, P., and Cooper, O.: Diurnal cycle of ozone throughout the troposphere over Frankfurt as measured by MOZAIC-IAGOS commercial aircraft, Elem. Sci. Anth., 4, 10.12952/journal.elementa.000129, 2016.

Wild, O. and Prather, M. J.: Global tropospheric ozone modeling: Quantifying errors due to grid resolution, J. Geophys. Res., 111, D11305, https://doi.org/10.1029/2005jd006605, 2006.

Yin, H., Lu, X., Sun, Y., Li, K., Gao, M., Zheng, B., and Liu, C.: Unprecedented decline in summertime surface ozone over eastern China in 2020 comparably attributable to anthropogenic emission reductions and meteorology, Environ. Res. Lett., 16, 124069, 10.1088/1748-9326/ac3e22, 2021.

Yu, K., Jacob, D. J., Fisher, J. A., Kim, P. S., Marais, E. A., Miller, C. C., Travis, K. R., Zhu, L., Yantosca, R. M., Sulprizio, M. P., Cohen, R. C., Dibb, J. E., Fried, A., Mikoviny, T., Ryerson, T. B., Wennberg, P. O., and Wisthaler, A.: Sensitivity to grid resolution in the ability of a chemical transport model to simulate observed oxidant chemistry under high-isoprene conditions, Atmos. Chem. Phys., 16, 4369–4378, https://doi.org/10.5194/acp-16-4369-2016, 2016.

Young, P. J., Naik, V., Fiore, A. M., Gaudel, A., Guo, J., Lin, M. Y., Neu, J. L., Parrish, D. D., Rieder, H. E., Schnell, J. L., Tilmes, S., Wild, O., Zhang, L., Ziemke, J. R., Brandt, J., Delcloo, A., Doherty, R. M., Geels, C., Hegglin, M. I., Hu, L., Im, U., Kumar, R., Luhar, A., Murray, L., Plummer, D., Rodriguez, J., Saiz-Lopez, A., Schultz, M. G., Woodhouse, M. T., and Zeng, G.: Tropospheric Ozone Assessment Report: Assessment of global-scale model performance for global and regional ozone distributions, variability, and trends, Elem. Sci. Anth., 6, p. 10, https://doi.org/10.1525/elementa.265, 2018.

**Comment [1-3]:** When doing the attributing analysis, taking aircraft for example, the authors fixed all the other emissions at 1995 level, and then varying the aircraft emissions. Since the world is experiencing significant emission changes for both developed regions (emission decreasing) and developing regions (emission

increasing), so how this method will affect the contribution from the aircraft for the ozone production without considering the realistic emissions in specific years?

**Response [1-3]: Thanks for your question. For estimating ozone trend from aircraft emissions, we conduct the simulation FixAircraft with aircraft emissions fixed at 1995 levels while other emissions are varied from 1995 to 2017. We then estimate the aircraft emission contributed ozone trend as the difference in trends between the BASE and FixAircraft simulations. In this method, we do have considered the aircraft ozone production with the realistic emissions from other sources in specific years. We have clarified in the text "In the third simulation (FixAircraft), we fix global aircraft emissions at 1995 levels, and use the difference in ozone trend between BASE and FixAircraft to estimate the contribution of aircraft emissions alone to ozone trends.".**

**Comment [1-4]:** L55-56: rephrase this sentence.

**Response [1-4]: We have modified the text as "Tropospheric ozone is produced chemically from anthropogenic and natural precursors, it is also transported from the stratosphere, and is removed by chemical loss and dry deposition."**

**Comment [1-5]:** L57: I feel change to "The ozone lifetime ranges/spans from xxx to ..." reads better. Just a suggestion.

**Response [1-5]: We have modified the text as "The ozone lifetime spans from hours in the polluted boundary layer to a few weeks in the free troposphere, sufficiently short that ozone distributions and trends are highly variable."**

**Comment [1-6]:** L70-79: I would encourage the authors to summary the findings from the IPCC report, instead of citing the whole paragraph for their own paper.

**Response [1-6]: Thanks for your suggestion. We have carefully evaluated and discussed this issue. Our co-author, Dr. Owen Cooper, was a contributing author to the tropospheric ozone assessment by IPCC. He and his colleagues were instructed to produce a very short statement that concisely and accurately summarized global tropospheric ozone trends. This was a difficult task given the regional variability of ozone trends around the world, and every word in the IPCC statement was scrutinized for accuracy. Given that IPCC has already produced an accurate and concise summary statement on global ozone trends, it's not possible for us to summarize the statement further as important information will be lost; in addition, we prefer not to change any of the words as they were carefully chosen for accuracy. For these reasons we suggest it's best to provide a direct quote from IPCC.**

**Comment [1-7]:** L202: The units for the trend should be "Tg yr-1" or "Tg decade-1".

**Response [1-7]: Corrected.**

**Comment [1-8]:** L214: change to "(Zheng et al., 2018)"

**Response [1-8]: Corrected.**

**Comment [1-9]:** Fig. 1: Explain the color indications for low panel

**Response [1-9]: Thanks for pointing this out. We have clarified in the text: "The lower panel shows the location of selected ozonesonde sites in 1995–2017 used in this study, grouped by six latitude bands with an interval of 30° as denoted by different colors."**

**Comment [1-10]:** Fig. 2: Explain the meanings in the bracket in left panels. The unit for the right panel is not accurate. Also reading the right panels for the trends of NOx CO and NMVOC, the authors can use the unit of "Tg/decade".

**Response [1-10]: Thanks for pointing this out. We have clarified in the figure, and added the following text in the caption of Fig.2 "The total global anthropogenic emission trends with $p$-value are shown in left panels."**

---

## Author Comment (AC2)

Dear Editor Dr. Leiming Zhang,

Thank you very much for handling our manuscript. Please find below our itemized responses to the reviewers' comments and a marked-up manuscript. We have addressed all the comments raised by the reviewer and incorporated them in the revised manuscript.

Thank you for your consideration.

Sincerely,
Haolin Wang et al.
* * *
Reviewer #2

**Comment [2-1]:** General comment: The research paper by Wang et al. shows decent work on analyzing multi-platform tropospheric ozone observations along with modelling study. The results of tropospheric ozone trends and emission-driven result via aircraft contributions are important for the research community. The model simulation was done with coarse grids, which could be improved, but the general results are solid. I would recommend publishing this work after addressing the following comments.

Response [2-1]: We thank the reviewer for the positive comments on our study. Below, we provide a point-by-point response to the reviewer's comments and summarize the changes that have been made in the revised manuscript.

**Comment [2-2]:** Figure 2: The trend of the aircraft emissions should also be included in the figures, not just CEDS results.

Response [2-2]: We have added the trend of the aircraft emissions in the Figure.

**Comment [2-3]:** L298-299: The simulation is on very coarse grids, i.e., $4° \times 5°$. Even for mid-latitude regions, the footprint of the model grid would be an area of 300 km $\times$ 500 km. How often is such a process needed (especially for ozonesondes)? A related question is, unlike stratospheric ozone, tropospheric ozone has more fine-scale structures (e.g., surface pollution, lightning, etc.). Could you please provide any comments on why not higher resolution simulation was used for this work?

Response [2-3]: We thank the reviewer for bringing this up. Referee #1 also raises this issue. To address both referees' comments, we have added the following discussion in Section 2.3 "We run the GEOS-Chem model at a horizontal resolution of 4° (latitude) $\times$ 5° (longitude), with 72 vertical layers extending from surface to 0.01 hPa. One-month model simulation at this resolution costs 4 hours with 48 CPUs (http://wiki.seas.harvard.edu/geos-chem/index.php/GEOS-Chem_13.3.0#1-month_benchmarks).
Yielding 33-year (including 10-year spin-up simulation) global simulation of ozone trends thus require computation time of more than 60 natural days. As such we do not use a finer resolution of 2°$\times$ 2.5° that would otherwise cost at least eight times as much computational time and resources as in this study. This relatively coarse resolution of 4°$\times$ 5° may limit the ability of the model to capture finer-scale ozone trends, in particular at near surface where ozone and its precursor has a short lifetime. Artificial mixing of surface ozone precursors in coarse model grids may lead to higher-than-actual ozone production efficiency and therefore positive ozone biases which may further influence trend analyses (Wild and Prather, 2006; Yu et al., 2016; Young et al, 2018; Yin et al., 2021). The limitation of model resolution, however, should be

**alleviated for ozone in the free troposphere, where ozone has longer chemical lifetime and should be better mixed than at near surface (Petetin et al., 2016). In light of this we do not use surface ozone observations for model evaluation, and mainly focus the trend analyses on above 950 hPa."**

Reference:

Petetin, H., Thouret, V., Athier, G., Blot, R., Boulanger, D., Cousin, J. M., Gaudel, A., Nédélec, P., and Cooper, O.: Diurnal cycle of ozone throughout the troposphere over Frankfurt as measured by MOZAIC-IAGOS commercial aircraft, Elem. Sci. Anth., 4, 10.12952/journal.elementa.000129, 2016.

Wild, O. and Prather, M. J.: Global tropospheric ozone modeling: Quantifying errors due to grid resolution, J. Geophys. Res., 111, D11305, https://doi.org/10.1029/2005jd006605, 2006.

Yin, H., Lu, X., Sun, Y., Li, K., Gao, M., Zheng, B., and Liu, C.: Unprecedented decline in summertime surface ozone over eastern China in 2020 comparably attributable to anthropogenic emission reductions and meteorology, Environ. Res. Lett., 16, 124069, 10.1088/1748-9326/ac3e22, 2021.

Yu, K., Jacob, D. J., Fisher, J. A., Kim, P. S., Marais, E. A., Miller, C. C., Travis, K. R., Zhu, L., Yantosca, R. M., Sulprizio, M. P., Cohen, R. C., Dibb, J. E., Fried, A., Mikoviny, T., Ryerson, T. B., Wennberg, P. O., and Wisthaler, A.: Sensitivity to grid resolution in the ability of a chemical transport model to simulate observed oxidant chemistry under high-isoprene conditions, Atmos. Chem. Phys., 16, 4369–4378, https://doi.org/10.5194/acp-16-4369-2016, 2016.

Young, P. J., Naik, V., Fiore, A. M., Gaudel, A., Guo, J., Lin, M. Y., Neu, J. L., Parrish, D. D., Rieder, H. E., Schnell, J. L., Tilmes, S., Wild, O., Zhang, L., Ziemke, J. R., Brandt, J., Delcloo, A., Doherty, R. M., Geels, C., Hegglin, M. I., Hu, L., Im, U., Kumar, R., Luhar, A., Murray, L., Plummer, D., Rodriguez, J., Saiz-Lopez, A., Schultz, M. G., Woodhouse, M. T., and Zeng, G.: Tropospheric Ozone Assessment Report: Assessment of global-scale model performance for global and regional ozone distributions, variability, and trends, Elem. Sci. Anth., 6, p. 10, https://doi.org/10.1525/elementa.265, 2018.

**Comment [2-4]:** L309: this seasonal bias cannot be seen in Figure 3. If this is provided in other figures or supplements, please indicate it properly.

**Response [2-4]: Thank you for pointing it out. We have further added the comparison of the simulated lower tropospheric ozone (950-800 hPa) with the IAGOS observation and the seasonal biases were shown in Table S1 in the Supplement Information. We now state in the text: "The modelled ozone is biased high in the tropical regions particularly in boreal autumn and winter (Table S1)."**

**Table S1.** Seasonal biases (ppbv) between observed and modeled lower tropospheric ozone (950-800 hPa) for 11 IAGOS regions from 1995-1999 to 2013-2017.

| Region | 1995-1999 | | | | 2013-2017 | | | |
|---|---|---|---|---|---|---|---|---|
| | MAM | JJA | SON | DJF | MAM | JJA | SON | DJF |
| East Asia | -7.6 | 9.3 | 1.6 | -7.4 | -10.5 | 4.6 | -0.4 | -10.1 |
| India | 13.6 | 8.1 | 15.3 | 16.2 | 4.9 | 6.4 | 2.8 | 4.8 |
| Southeast Asia | 6.9 | 11.3 | 14.3 | 11.2 | 1.1 | 3.1 | 11.4 | 5.6 |
| Persian Gulf | 5.7 | 12.5 | 19.7 | 4.3 | 0.4 | 9.5 | 13.7 | -0.1 |
| Malaysia/Indonesia | 7.5 | 9.3 | 24.5 | 14.5 | 0.8 | -1.3 | 0.4 | 4.6 |
| Gulf of Guinea | 10.3 | 5.6 | 10.2 | 18.2 | -7.6 | -2.4 | 1.3 | -1.2 |
| Northern South America | 3.8 | 0.8 | 1.8 | 7.1 | -12.5 | -6.2 | 2.3 | 5.1 |

| | | | | | | | | |
|---|---|---|---|---|---|---|---|---|
| Europe | -3.6 | 8.7 | 1.8 | -4.7 | -4.1 | 3.9 | 0.9 | -6.6 |
| Eastern North America | 2.0 | 13.0 | 10.1 | -1.5 | -1.8 | 5.8 | 6.7 | -2.1 |
| Southeast US | 7.6 | 15.7 | 10.4 | 2.8 | -3.5 | 4.5 | 3.1 | 0.3 |
| Western North America | -3.2 | 13.0 | 3.4 | -4.4 | -9.9 | -4.1 | -6.1 | -9.5 |

**Comment [2-5]:** L326-329: The previous section mentioned the current work (simulation/analysis) already removed contributions from STE. I.e., L272-275 (remove data points with ozone higher than 125 ppbv at altitudes higher than 500 hPa). Is the same filter been applied to Figure 3 (or just the trend part)? If yes, this argument of STE should be clarified carefully. If not, please provide reasoning why not.

**Response [2-5]: Yes. We remove data points with ozone higher than 125 ppbv at altitudes higher than 500 hPa for both model evaluation of absolute values and trends (Figs.3-7). This is to exclude the effect of episodic stratospheric intrusion which is hard for the model to capture due to the dilution of these intrusion effects in the model grid. This only excludes episodic stratospheric intrusions, while stratospheric ozone contribution through large-scale general circulation is still considered in the model evaluation. We have clarified that argument in L326-329 is referred to an earlier version of the model "The latest comprehensive evaluation of the global tropospheric ozone simulation using the version 10.1 of the model (Hu et al., 2017) found small low ozone biases compared to ozonesonde observations in the northern extratropical and polar regions, which were attributed to the underestimation of stratosphere-troposphere ozone exchange (STE) flux in that version of the model".**

**Comment [2-6]:** Figure 4: The bias between model and ozonesondes is larger and "uniform" (from the surface up to near tropopause) in Polar Regions, when compared to lower latitudes results. Do you think there could be other systematic bias in the model causing such a feature? The sites for Polar Regions are very sparse (only three Canadian sites in the Arctic, and only two Antarctic sites), but all five sites show consistent feature in term of bias (Fig. S2). I.e., it is not an averaging issue.

**Response [2-6]: Thank you for pointing it out. This is most likely due to updates on halogen chemistry due to a decrease in ozone production (the sink of NOx from formation and hydrolysis of $ClNO_3$ and $BrNO_3$) and an increase in ozone chemical loss (catalytic cycles involving iodine and bromine) (Wang et al., 2021). We have added the following text "The scientific updates since the version 10.1 (https://geos-chem.seas.harvard.edu/geos-new-developments) …, in particular updated halogen chemistry further decreases surface ozone at high-latitude regions (Wang et al., 2021)."**

References

Wang, X., Jacob, D. J., Downs, W., Zhai, S., Zhu, L., Shah, V., Holmes, C. D., Sherwen, T., Alexander, B., Evans, M. J., Eastham, S. D., Neuman, J. A., Veres, P. R., Koenig, T. K., Volkamer, R., Huey, L. G., Bannan, T. J., Percival, C. J., Lee, B. H., and Thornton, J. A.: Global tropospheric halogen (Cl, Br, I) chemistry and its impact on oxidants, Atmos. Chem. Phys., 21, 13973-13996, 10.5194/acp-21-13973-2021, 2021.

**Comment [2-7]:** L342-343: Please clarify that the numbers provided in this paragraph here are only for IAGOS. I saw a discussion of modelled results later (e.g., L386-391).

**Response [2-7]: We have clarified in the text.**

**Comment [2-8]:** L344-346: I am not challenging the authors about this seasonal difference, i.e., "largely driven by boreal winter" or "driven by ozone decreases in the summer". But, the figures provided here are only annual trends. Please provide evidence to support the argument. If this is using figures in the later part, please give some indications.

**Response [2-8]: We have added a figure and clarified in the text: "In comparison, the lower tropospheric trends of the 50th percentile ozone in developed regions (Europe, North America) over the northern mid-latitudes are much smaller by up to 1.8 ppbv decade$^{-1}$, which is largely driven by boreal autumn and winter with ~1.2 ppbv decade$^{-1}$ on average (Fig.S4). There are small negative trends in the annual 50th percentile in the lower troposphere above North America driven by ozone decreases in the summer (Fig.S4) (Cooper et al., 2012; Simon et al., 2015; Gaudel et al., 2020)."**

[Figure]

Figure S4. Same as Figure 5 but for seasonal trends of the 50th percentiles of IAGOS observed ozone.

**Comment [2-9]:** L371-372: Two issues here. First, Payerne's sampling rate is not any close to 4 profiles/month. Table 1 says it is 12 profiles/month. Also, four profiles/month is not something "unusual", i.e., only 6 out of 27 sites have a sampling rate >= 5. So, with current evidence, I could not support that these negative trends detected at some sites are simply due to their low sampling rate at 4. One must find stronger support.

**Response [2-9]: Thank you for pointing it out. We have rephrased the text as "In Europe and North America, observed trends are mostly positive, while three sites (Payerne, Legionowo, and Boulder) show inconsistently negative trends of -0.5~-0.6 ppbv decade$^{-1}$ that are in contrast to IAGOS observations (0.8~1.7 ppbv decade$^{-1}$) and trends at the other nearby sites (0.3-2.1 ppbv decade$^{-1}$). Increasing the sampling frequency (i.e. to 18 profiles month$^{-1}$ according to Chang et al. (2020) would be helpful to reconcile the ozone trend estimate at adjacent ozonesonde sties, but we do not exclude the possibility that tropospheric ozone trends can still be variable even at adjacent locations."**

**Comment [2-10]:** L424-429: This part along with Figure 9 show key information here. Figure 9 panel (b) shows the "Aircraft-drive trends", while it is a bit confusing which part it will contribute to panel (a). The Aircraft emission is much lower in terms of the tropospheric ozone burden, as described in the previous section. Some better description is needed.

**Response [2-10]: The tropospheric ozone burden driven by aircraft emissions in Figure 9 panel (a) is included in the green shadings, which estimate the tropospheric ozone burden contributed by the anthropogenic emission (including aircraft emission) changes relative to the 1995 level. We prefer not to add an orange shading for aircraft emission contributing ozone burden because that will make the green bars inconsistent between panels (a) and (b). We now clarify in the figure caption: "The green shadings thus estimate the tropospheric ozone burden contributed by the anthropogenic emission (including aircraft emission) changes relative to the 1995 level."**

**Comment [2-11]:** L467-476 & Figures 8 and 9: In terms of trends, the sharp peak in GEOS-Chem simulation in 1998 looks interesting. Authors attributed this to ENSO. However, such a feature is only captured by only a few CMIP6 models, e.g., CESM2-WACCM. This feature is more prominent for GEOS-Chem than any other modelled results. For GEOS-Chem, the peak value in 1998 is only a few Tg Ozone less than the values in 2017. Without this peak, the trend of GEOS-Chem would be more clear and more significant. Any comments or explanations on this feature? From Figure 9, this 1998 peak is not new "emission-driven" at least.

**Response [2-11]: Thank you for pointing this feature out. We have added the following text as discussion: "This El Niño driven ozone peak in 1998 is more prominent in GEOS-Chem than most of the CMIP6 models, very likely because El Niño driven shift in weather conditions and transport pattern is better reflected in MERRA2 re-analyses data used to drive our GEOS-Chem model, compared to climate fields simulated by CMIP6 climate-chemistry models without nudging to observed sea-surface temperature."**

**Comment [2-12]:** L484-485: I could not support this description about stratospheric ozone being observed having a recovery. Most recent works only show some level of the signal of possible recovery from observations (e.g., Weber et al. 2022; 10.5194/acp-22-6843-2022). One thing the community agreed on is that the Montreal Protocol levelled off the decreasing trend. But, no solid observational-based stratospheric ozone recovery could be claimed yet (e.g., see Weber's results). Please wording this part carefully.

**Response [2-12]: Thank you for pointing it out. We now state in the text: "Our GEOS-Chem model by implementing the time-resolved surface concentrations of ozone-depleting-species as boundary conditions shows a moderate increasing trend in total stratospheric ozone burden from 1995 to 2017 (Fig.S7), consistent with the observations suggesting a leveling off of declining trends in stratospheric ozone after the Montreal Protocol (Solomon et al., 2016; Weber et al. 2022) and with other modeling studies (Griffiths et al., 2020)."**

**Comment [2-13]:** L495: The STE trends in Figure S7 are global average results, not for high latitudes. Trends for different latitude bands should be generated to show the feature and support this argument.

**Response [2-13]: Unfortunately, both methods (residual budget and vertical ozone flux at 100 hPa) are not able to calculate the trend of STE for different latitude bands. In future we hope to use tagged $O_3$ simulation to track the $O_3$ from the stratosphere and to better understand the trends in STE. We now state in the text: "However, both methods are not applicable to derive STE trends at different latitude bands. More work is required to evaluate the trends in STE flux and to explore the driving factors."**

**Comment [2-14]:** Figure 12: Well, this figure is confusing. I think panel (a) is only a tropospheric column, but why do panels (b) and (c) show results up to 0 hPa? If so, I would assume they are total columns. Anyway, Figure 12b shows that even for the most significant increasing layer (i.e., tropic above 200 hPa), the amount of increase is only at a level of 0.2 DU. Total column ozone is normally at a level of 300 DU. Any comments on the significance of these changes to radiation via total ozone contribution?

**Response [2-14]: Sorry for the confusion. All the numbers are for tropospheric ozone column. We have revised the figure accordingly to avoid the confusion.**

**Comment [2-15]:** L514-524: The description here is strange to me (i.e., it says its total ozone radiative impacts, while Figure 13 says its tropospheric ozone radiative impacts). I think this paragraph is talking about tropospheric ozone radiative impact, not total. Please be specific. Note that tropospheric ozone is only about 10% of total ozone.

**Response [2-15]: Sorry for the confusion. We have clarified that the discussions are all about tropospheric ozone radiative impact. Here "total" refers to the combination of longwave and shortwave radiative impact.**

**Comment [2-16]:** L547-549: Well, 1995-2017 results from GEOS-Chem might look slightly "better" when compared with CMIP6. However, it is more important to show comparison results for the same time window. For 1995-2014, GEOS-Chem's trend is only 0.2 Tg year-1. This key result should be included in the conclusion.

**Response [2-16]: We have rephrased the text as "GEOS-Chem estimates an increasing trend in global tropospheric ozone burden of 0.2 Tg year$^{-1}$ in 1995–2014 (0.4 Tg year$^{-1}$ in 1995–2017), compared to the CMIP6 model ensemble of 0.4 to 1.3 Tg year$^{-1}$ in 1995–2014."**

**Comment [2-17]:** L566-568: Could also include % changes in the description. The absolute number here is important but less informative for the conclusion.

**Response [2-17]: Thank you for the suggestion. We have added the percentage change, as shown below:**

   **"We estimate a global mean tropospheric ozone total radiative impact of 18.5 mW m$^{-2}$ in 2013–2017 compared to 1995–1999 level, with an increase by ~1.2%, but we suggest the true radiative impacts should be larger as our simulation underestimates the overall tropospheric ozone trends from 1995-2017."**

**Comment [2-18]:** L126: the notation of accuracy looks very strange, please double check.

**Response [2-18]: We thank the reviewer for this valuable comment. We have updated the notation of accuracy.**

**Comment [2-19]:** Figure 10: unit of the y-axis is missing.

**Response [2-19]: Corrected.**

**Comment [2-20]:** Figures 12b and 12c: unit of the y-axis is missing.

**Response [2-20]: Corrected**

**Comment [2-21]:** L458: it is not an ozone trend, but a tropospheric ozone trend. Similar to other parts in this work, please be specific.

**Response [2-21]: We have corrected throughout the text.**

**Comment [2-22]:** L469: STE was defined in the previous section.

**Response [2-22]: Corrected.**

**Comment [2-23]:** Figures S2 and S8: unit of the y-axis is missing.

**Response [2-23]: Corrected.**

**Comment [2-24]:** L539: tropospheric ozone increases.

**Response [2-24]: Corrected.**